# RANKL regulates male reproductive function

Martin Blomberg Jensen [1,2✉], Christine Hjorth Andreassen [1,7], Anne Jørgensen [3,7], John Erik Nielsen[3],
Li Juel Mortensen[1,2], Ida Marie Boisen[1,2], Peter Schwarz [4,5], Jorma Toppari[6], Roland Baron [2],
Beate Lanske[2] & Anders Juul[3,5]

Infertile men have few treatment options. Here, we demonstrate that the transmembrane receptor activator of NF-kB ligand (RANKL) signaling system is active in mouse and human testis. RANKL is highly expressed in Sertoli cells and signals through RANK, expressed in most germ cells, whereas the RANKL-inhibitor osteoprotegerin (OPG) is expressed in germ and peritubular cells. OPG treatment increases wild-type mouse sperm counts, and mice with global or Sertoli-specific genetic suppression of Rankl have increased male fertility and sperm counts. Moreover, RANKL levels in seminal fluid are high and distinguishes normal from infertile men with higher specificity than total sperm count. In infertile men, one dose of Denosumab decreases RANKL seminal fluid concentration and increases serum Inhibin-B and anti-Müllerian-hormone levels, but semen quality only in a subgroup. This translational study suggests that RANKL is a regulator of male reproductive function, however, predictive bio-markers for treatment-outcome requires further investigation in placebo-controlled studies.

[1] Group of Skeletal, Mineral and Gonadal Endocrinology, Department of Growth and Reproduction, Rigshospitalet, University of Copenhagen, Copenhagen, Denmark. [2] Division of Bone and Mineral Research, HSDM/HMS, Harvard University, Boston, MA, USA. [3] Department of Growth and Reproduction and International Center for Research and Research Training in Endocrine Disruption of Male Reproduction and Child Health (EDMaRC), Rigshospitalet, University of Copenhagen, Copenhagen, Denmark. [4] Department of Endocrinology, Rigshospitalet, University of Copenhagen, Copenhagen, Denmark. [5] Faculty of Health and Medical Sciences, University of Copenhagen, Copenhagen, Denmark. [6] Institute of Biomedicine, Research Centre for Integrated Physiology and Pharmacology, and Centre for Population Health Research, University of Turku, and Department of Pediatrics, Turku University Hospital, Turku, Finland. [7] These authors contributed equally: Christine Hjorth Andreassen, Anne Jørgensen. ✉email: blombergjensen@gmail.com

nfertility is a frequent problem affecting between 7–26% of all couples globally[1,2]. Impaired semen quality is responsible for approximately 50% of all cases, but no treatment options to improve semen quality exist for most infertile men except for those with hypogonadotropic hypogonadism and varicocele[3,4]. The mechanisms underlying reduced sperm concentration, motility, and aberrant sperm morphology are not completely understood, but Sertoli cell function plays an essential role[5]. Spermatogenesis is strongly influenced by testosterone produced by testicular Leydig cells under the control of luteinizing hormone (LH), whereas anti-Müllerian hormone (AMH) and Inhibin B secretion from Sertoli cells are stimulated by follicle stimulating hormone (FSH)[6]. There exists an endocrine crosslink between the gonads and bone as sex steroids are potent regulators of skeletal function[7], while influence of FSH, activins, and inhibins on skeletal function is more controversial[8,9]. Bone specific proteins for instance osteocalcin has been proposed to stimulate testosterone and sperm production, and vitamin D induces sperm motility and increases live birth rates in oligospermic men[10–13]. Studies in human testis and vitamin D receptor (Vdr) knockout mice have shown that several bone factors such as Runt-related transcription factor 2 (RUNX2), Osterix, and osteocalcin are expressed both in normal testis as well as in testicular germ cell tumours (TGCTs)[14,15]. This highlights that some of these putative "endocrine bone factors" may act locally in the gonad and influence testicular function. One of these factors is the receptor activator of NF-κB ligand (RANKL). The RANKL system is an essential regulator of bone resorption that comprises three components. RANKL, a transmembrane ligand that upon binding to its receptor RANK on a neighboring cell activates NF-κB and regulates cell cycle i.e., proliferation, differentiation, and apoptosis[16,17]. The transmembrane RANKL resides in bone-forming cells and activates RANK in osteoclast precursor cells and thereby induces osteoclastogenesis and promotes bone resorption[17]. Osteoprotegerin (OPG) is a secreted decoy receptor that binds RANKL and blocks activation of RANK[18] and prevents osteoclast differentiation and activation[19]. RANKL can also be found in a soluble form in serum, suggesting a putative endocrine function and non-skeletal actions for instance in bone and muscle health, menstrual cycle, and glucose homeostasis[9,20–24]. Moreover, a recent study showed how complex RANKL signaling is as it was suggested that reverse RANKL signaling occurs when the receptor RANK becomes soluble and binds to RANKL thereby inducing downstream signaling in the RANKL expressing cell[25]. Here, we show that RANKL, RANK, and OPG are expressed in different cell types in the testis, suggesting a yet unrecognized regulatory role of spermatogenesis. These findings may ultimately be of clinical relevance since an FDA/EMA approved specific RANKL inhibitor, Denosumab, is in use to treat osteoporosis in women and men[26]. Initially, OPG was used clinically but discontinued due to off-target inhibition of tumour necrosis factor-related apoptosis-inducing ligand (TRAIL) and a short half-life. Denosumab was developed to avoid off-target effects and has a longer half-life than OPG, which imply that patients only require injections every six months for treatment of osteoporosis[27]. Here, we show that the male gonads share the RANKL/RANK/OPG signaling system with the bone and immune system; and by using exogenous and genetic repression of RANKL systemically and specifically in Sertoli cells in mice as well as descriptive and functional human studies we provide insights into the relevance of RANKL as a modifiable regulator of male reproductive function.

## Results

### RANKL, RANK, and OPG are expressed in the testis of mice and inhibition of RANKL increases testicular weight and sperm count. RANKL, RANK, and OPG were detected in the mouse testis and epididymis at 16–18 weeks of age (Fig. 1a–d;

Supplementary Fig. 1a–c). RANKL was strongly expressed in the cytoplasm/membrane of mature Sertoli cells co-expressing SOX9 in the nuclei (Fig. 1a). RANKL expression was also detected in the cytoplasm of most spermatocytes and some spermatids but not in spermatogonia (Fig. 1a, d; Supplementary Fig. 1a). RANK was expressed in the cytoplasm and membrane of the VASA-positive germ cells, particularly spermatogonia and spermatids and in some spermatocytes (Fig. 1b, d; Supplementary Fig. 1a). OPG was expressed in the cytoplasm of most spermatogonia, the junction between peritubular cells and spermatogonia, spermatids, and some peritubular cells (Fig. 1c, d). In the epididymis, RANKL was markedly expressed in the caput, while being undetectable or with low RANKL expression in corpus and cauda epididymis. RANK and OPG were also expressed in the epididymis and the expression varied in the different segments (Supplementary Fig. 1c). To manipulate gonadal cell–cell signaling of RANKL, the RANKL inhibitor OPG (1 mg/kg) was injected twice a week for two or five weeks into C57BL/6 mice. OPG treatment for two weeks significantly increased testicular weight and germ cell epithelial height of seminiferous tubules at stage VII–VIII when compared with vehicle-treated mice (Fig. 1e; Supplementary Fig. 2a). Two weeks of OPG treatment also increased sperm counts in cauda epididymis compared with vehicle-treated mice (32 vs. 21 million spermatozoa, $p < 0.05$) (Fig. 1e). The rapid stimulatory effect was surprising and could be the result of reduced apoptosis since the duration of spermatogenesis in mice is 35 days. Neither OPG nor OPG-FC (osteoprotegerin–immunoglobulin Fc segment complex) treatment for 5 weeks resulted in increased testicular weight or sperm count suggesting compensation or a transient anti-apoptotic effect (Fig. 1f). RANKL injected into C57BL/6 mice twice a week for 2 weeks also did not affect testicular weight, although epididymal weight was reduced compared with OPG-treated mice (Supplementary Fig. 2b).

### High sperm counts and increased fertility in mice with RANKL-deficiency. To determine the tissue and cell-specific function of RANKL inhibition in vivo we crossed $Rankl^{fl/fl}$ mice with anti-müllerian hormone (Amh); Cre Tg mice to generate a Sertoli cell-specific RANKL-deficient mouse line. This mouse was used for comparison with a global RANKL-deficient model established by crossing $Rankl^{fl/fl}$ with DEAD-Box Helicase 4 (Vasa); Cre Tg mice (Supplementary Methods 1). Vasa; Cre Tg mice are generally used to create germ cell-specific loss. However, Cre activity is global when the allele is either inherited from the mother, or when old males are used for breeding (Inf. S1). Genotyping was performed and further validated by backcrossing the mutant mice with wild-type (WT) mice, by crossing the RANKL-deficient lines with Rosa-reporter mice showing global expression exclusively in the Vasa;Cre positive mice, and Sertoli cell-specific activity in Amh;Cre positive mice with no activity in floxed mice (Fig. 2a; Supplementary Methods 1). Phenotypic characteristics, including loss of tooth eruption, lactation deficit, and increased bone mass in the Vasa;Cre model were consistent with the expectations of a global RANKL knockout model[19]. An increase in bone mass was evident following a rough but systematic evaluation of tibial bone mass (Fig. S3a). An assessment of the breeding schemes showed that some offspring of female Vasa-Cre mice died due to lactation deficiency (Supplementary Table 10). Moreover, a fraction of offspring from both female and male Vasa-Cre mice suffered from loss of tooth eruption (Supplementary Table 11). Littermates ($Rankl^{fl/fl}$ mice) without Cre activity were used as controls. Rankl expression in the testis varied considerably in $Rankl^{fl/fl}$ mice because 50% had heterozygous loss of Rankl, which is a known unpreventable side-effect when using germ cell-specific Cre mouse lines (Fig. 2b). However,

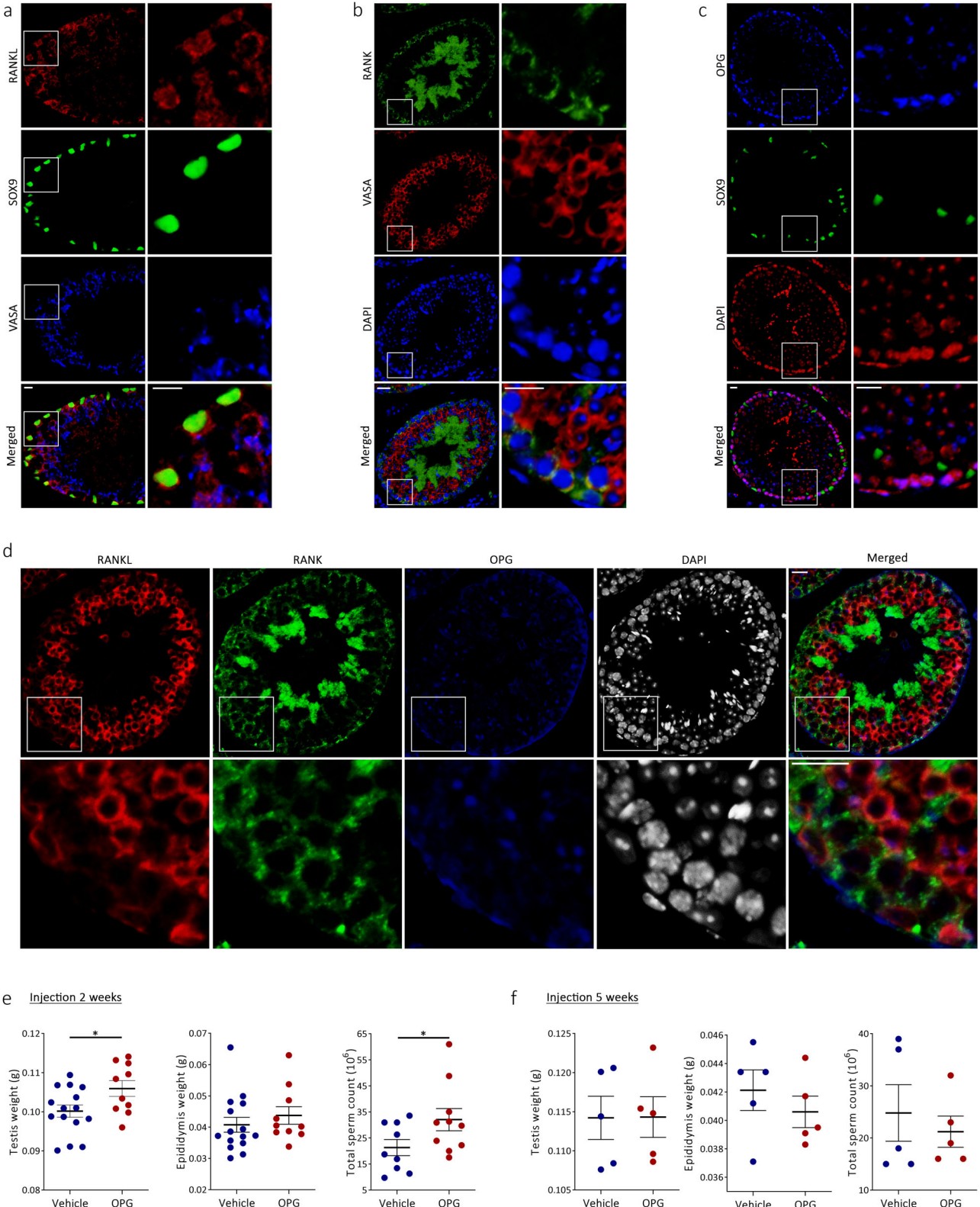

**Fig. 1 Gonadal RANKL signaling and influence of OPG in mice. a** Triple immunofluorescence with RANKL (sc-9073, red), SOX9 (Sertoli cell marker, green), and VASA (germ cell marker, blue). **b** Double immunofluorescence with RANK (HPA0277728, green), VASA (red), and DAPI (blue), or **c** OPG (sc-21038, blue), SOX9 (green), and DAPI (red). **d** Triple immunofluorescence with RANKL (sc-9073, red), RANK (HPA0277728, green), OPG (sc-21038, blue), and DAPI (gray) (all IF in 16-week-old WT mice). White insert marks high magnification area. **e, f** Phenotypic effects of OPG (1 mg/kg, red) or vehicle (blue) injected twice weekly for 2 or 5 weeks on testicular and epididymal weight and total epididymal sperm count in 10-week-old mice, (**e**, left panel; $p = 0.03$, right panel; $p = 0.04$). Note, $y$-axes do not intersect $x$-axes at 0 (**e, f**). Scale bars correspond to 25 μm (**a–d**). Data are presented individually and as mean ± SEM, with $n$ (vehicle/OPG) = 15/10 (left and middle panels, **e**) $n$ (vehicle/OPG) = 9/10 (right panel, **e**), and $n$ (vehicle/OPG) = 5/5 (all panels, **f**). Statistical test: two-sided Student's $t$ test (**e**, left and right panels) with *$p < 0.05$.

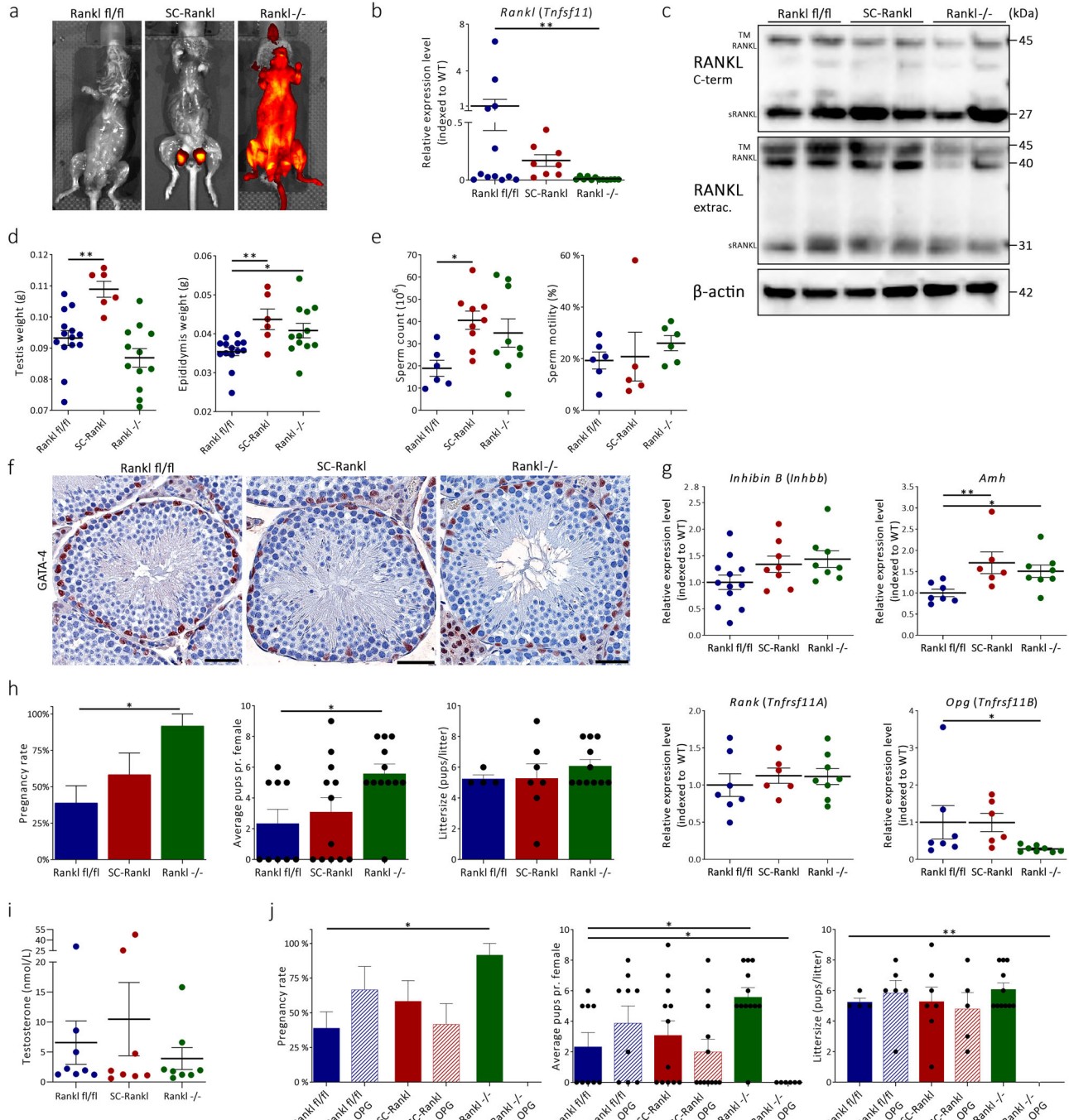

**Fig. 2 Phenotypic effects of systemic versus Sertoli cell-specific suppression of RANKL. a** Rosa reporter expression in *Rankl* fl/fl, Sertoli specific (AMHCre;Ranklfl/fl, SC-*Rankl*), and global (VasaCre;Ranklfl/fl, *Rankl* −/−) RANKL-deficient mice. **b, c** Expression of RANKL at mRNA and protein level in *Rankl* fl/fl (blue), SC-*Rankl* (red), and *Rankl* −/− (green) mice (**b**, fl/fl vs. SC: *p* = 0.923, fl/fl vs. −/−: *p* = 0.004). **d** Testicular and epididymal weight of *Rankl* fl/fl, SC-*Rankl*, and *Rankl* −/− mice, (left panel; *p* = 0.003, *p* = 0.159, right panel; *p* = 0.008, *p* = 0.033). **e** Sperm count and ABC sperm motility of *Rankl* fl/fl, SC-*Rankl*, and *Rankl* −/− mice, (left panel; *p* = 0.013, *p* = 0.107). **f** Morphology and expression of Sertoli cell marker GATA-4 in testis of *Rankl* fl/fl, SC-*Rankl*, and *Rankl* −/− mice. Counterstaining with Mayer's hematoxylin. Scale bars correspond to 50 μm. **g** Gene expression of *Inhibin B*, *Amh*, *Rank*, and *Opg* in *Rankl* fl/fl, SC-*Rankl*, and *Rankl* −/− mice, (top right panel; *p* = 0.01, *p* = 0.02, bottom right panel; *p* = 0.07, *p* = 0.04). **h** Fertility study of *Rankl* fl/fl, SC-*Rankl*, and *Rankl* −/− male mice exposed to three healthy females for 5 days, (left panel; *p* = 0.02, middle panel; *p* = 0.01). **i** Serum testosterone (nmol/L) in *Rankl* fl/fl, SC-*Rankl*, and *Rankl* −/− mice. **j** Fertility study of *Rankl* fl/fl, SC-*Rankl*, and *Rankl* −/− mice after OPG treatment (1 mg/kg x 2 weekly for 2 weeks), (left panel; *p* = 0.02, middle panel; *p* = 0.02, *p* = 0.01, right panel; *p* = 0.005). The OPG-treated *Rankl* −/− mice did not give birth to pups. Note, y-axes do not intersect x-axes at 0 (**d**). Data are presented individually and as mean ± SEM, with *n* (*Rankl* fl/fl ∕ SC-*Rankl* ∕ *Rankl* −/−) = 12/ 8/12 (**b**), *n* (= 14/6/12 (**d**), *n* = 6/9/9 (left panel, **e**), *n* = 6/5/6 (right panel, **e**), *n* = 4–6 (**f**), *n* = 12/8/8 (top left panel, **g**), *n* = 7/6/8 (top right, bottom left and right panels, **g**), *n* (male/female) = 6/18 (*Rankl* fl/fl) and *n* = 4/12 (SC-*Rankl*) and *n* = 4/12 (*Rankl* −/−) for all panels (**h**), *n* = 8 (**i**), *n* (male/ female) = 3/9 (*Rankl* fl/fl + OPG) and *n* = 4/12 (SC-*Rankl* + OPG) and *n* = 2/6 (*Rankl* −/− + OPG) (**j**). Statistical tests: one-way analysis of variance (ANOVA) with Dunnett's test to adjust for multiple comparisons (**b, d, e, g, h** (middle panel), **j** (middle and right panels)) or Pearson chi square test (**h** (left panel), **j** (left panel)) with *\*p* < 0.05, *\*\*p* < 0.01.

*Rankl* expression in control mice was significantly higher than in global RANKL-deficient mice at transcriptional and protein level (Fig. 2b, c). RANKL expression in Sertoli cell-specific RANKL-deficient mice did not differ significantly from *Rankl*$^{fl/fl}$ control mice (Fig. 2b, c).

The reproductive phenotype was evaluated in both models at the age of 16–17 weeks. Testicular weight was significantly higher exclusively in the Sertoli cell-specific RANKL-deficient mice compared with *Rankl*$^{fl/fl}$ mice, while epididymal weight was higher in both *Rankl*-deficient models compared with controls (Fig. 2d). Sperm count tended to be higher although not statistically significant in global RANKL-deficient compared with *Rankl*$^{fl/fl}$ mice (18 vs. 35 million sperm/ml, $p = 0.11$), whereas sperm count was significantly higher in the Sertoli cell-deficient model compared with *Rankl*$^{fl/fl}$ mice (18 vs. 40 million sperm/ml, $p = 0.01$) (Fig. 2e). Sperm motility was not significantly different in the two RANKL-deficient models compared with *Rankl*$^{fl/fl}$ mice (Fig. 2e). Histologically, no observable differences in spermatogenesis, number of peritubular or Leydig cells were found between the RANKL-deficient models and control mice (Fig. 2f; Supplementary Fig. 3c, d). However, seminiferous tubule diameter was higher in mice with global RANKL-deficiency compared with *Rankl*$^{fl/fl}$ mice ($p < 0.05$) (Supplementary Fig. S3b). In accordance, the expression level of *Inhibin B* tended to be higher albeit not significantly different after adjustment for multiple comparisons in mice with global or Sertoli cell specific RANKL-deficiency compared with *Rankl*$^{fl/fl}$ mice ($\Delta$*Inhibin B*: 44%, $p = 0.06$ and $\Delta$Inhibin B: 34%, $p = 0.14$, respectively) (Fig. 2g). However, *Amh* expression was significantly higher in both global and Sertoli specific RANKL-deficient models compared with control mice ($p < 0.05$ and $p < 0.01$, respectively) (Fig. 2g). *Rank* was not differentially expressed in RANKL-deficient models, but *Opg* was significantly lower in mice with global RANKL-deficiency compared with *Rankl*$^{fl/fl}$ mice ($p < 0.05$) (Fig. 2g). WT female mice mated with global RANKL-deficient males had a higher pregnancy rate than control mice (Fig. 2h). Also, the average number of healthy pups per mated female was significantly higher, while litter size did not differ when comparing global RANKL-deficient and *Rankl*$^{fl/fl}$ mice ($p < 0.05$) (Fig. 2h). No significant difference in pregnancy rate, average number of pups, or litter size was observed in the Sertoli cell-specific RANKL-deficient mating couples compared with *Rankl*$^{fl/fl}$ mice (Fig. 2h). The differences in fertility rates between genotypes were in part mirrored by number of days with verified plugs in exposed females (*Rankl* fl/fl 39%, *Rankl* fl/fl + OPG 56%, SC-*Rankl* 58%, SC-*Rankl* + OPG 67%, *Rankl* −/− 83%, and *Rankl* −/− + OPG 50%). However, the high frequency of verified plugs in OPG-treated *Rankl* −/− that produced no litters and the observed pregnancies without verified plugs in 15% of all females challenge the conclusions drawn from this observation. Moreover, serum levels of testosterone did not differ significantly between RANKL-deficient mice and controls, which implies that altered sexual activity alone is unlikely to explain the difference in pregnancy rates (Fig. 2i). Tendencies towards higher pregnancy rate (39 vs. 67% healthy pregnancies), average number of pups per female (2.1 vs. 3.9), and litter size (5.3 vs. 5.8) were observed in OPG-treated *Rankl*$^{fl/fl}$ mice although not statistically significant different from vehicle-treated *Rankl*$^{fl/fl}$ (Fig. 2j). OPG treatment of Sertoli cell RANKL-deficient mice did not increase fertility, while OPG treatment in global RANKL-deficient mice suppressed fertility and caused no pregnancies and litter sizes were zero in all six females (92% vs 0%; 6.1 vs 0.0 $p = 0.056$ and $p = 0.005$, respectively) (Fig. 2j).

**RANKL, RANK, and OPG are expressed in the human testis and regulate germ cell apoptosis.** Human testis specimens with invasive cancer cells were discarded, while samples with all stages of spermatogenesis (normal testis), dysgenetic samples with varying degrees of Sertoli-cell-only, spermatogenic arrest, and tubules possibly containing germ cell neoplasia in situ (GCNIS) were included. Expressions of *RANKL*, *RANK*, and *OPG* were found in the non-malignant testicular tissue (Fig. 3a; Supplementary Fig. 4a). All three isoforms of RANKL were expressed in the human testis including the transmembrane and soluble form of the protein (Supplementary Fig. 4c; Supplementary Table 1). The expression pattern of RANKL, RANK, and OPG was evaluated in formalin-fixed human testis to extrapolate the findings from mice to humans (Supplementary Fig. 5a–e). By using antibodies targeting the transmembrane or extracellular domain we found cytoplasmic expression of RANKL in the SOX9-positive Sertoli cells and in some of the germ cells on the luminal site of the blood-testis barrier (Supplementary Fig. 4b; Supplementary Fig. 5a, d–e; Supplementary Table 1). RANKL expression in germ cells was detected primarily in some spermatocytes and spermatids but not in spermatogonia (Supplementary Fig. 4b; Supplementary Fig. 5a, d–e; Table S1). RANK was expressed in the cytoplasm/membrane of the germ cells with the most prominent expression in spermatogonia and spermatids (Supplementary Fig. 4b, 5b–e; Supplementary Table 1). OPG was expressed in spermatogonia and less frequently in peritubular cells or at the border between the peritubular cells and spermatogonia and in a fraction of spermatids (Supplementary Fig. 4b, 5c–e; Supplementary Table 1). The same antibodies were also applied on human testis fixed in modified Stieve's solution which improve preservation of testicular morphology. Here, RANKL expression was also predominantly expressed in the cytoplasm/membrane of the Sertoli cells and in some spermatocytes and spermatids (Fig. 3b, e; Supplementary Table 1). RANK was expressed in the cytoplasm/membrane of the germ cells with the most prominent expression in spermatogonia and spermatids (Fig. 3c, e; Supplementary Table 1), while OPG was strongly expressed in spermatogonia, some of the peritubular cells, the border between the peritubular cells and spermatogonia and in many spermatocytes and spermatids (Fig. 3d, e; Supplementary Table 1). The soluble isoform of RANKL (sRANKL, observed at 27–31 kDa) and the transmembrane isoform (45 kDa) were both expressed in the testis (Supplementary Fig. 4c). Both isoforms of RANKL were in addition to RANK and OPG robustly expressed in samples with normal spermatogenesis. Analyses of the expression in Sertoli-cell-only tubules from four patients showed virtually no detectable RANK expression, but some cytoplasmic RANKL expression in the Sertoli cells characterized by SOX9 and Vimentin expression. Noteworthy, OPG expression was detected in the cytoplasm of the Sertoli cells and was thus strikingly different from the expression observed in testicular samples that contained seminiferous tubules with "normal" Sertoli cells and germ cells, where OPG was expressed in peritubular and germ cells (Supplementary Fig. 4d). Human testicular tissue was cultured ex vivo to preserve the functional integrity and to support continued cell–cell signaling in order to investigate effects of RANKL-inhibition on germ cell apoptosis. Most of the included specimens contained tubules with spermatogenic arrest or some atrophy (Supplementary Fig. 6a). Despite the low number of tubules with complete spermatogenesis, treatment with Denosumab or OPG for 48 h reduced the number of apoptotic cells (assessed by number of cleaved PARP$^+$ cells per area) by 30% compared with vehicle control ($p < 0.05$) (Fig. 3f, g). Denosumab or OPG treatment did not significantly affect germ cell proliferation (determined by number of BrdU$^+$ cells per area), although a tendency towards an

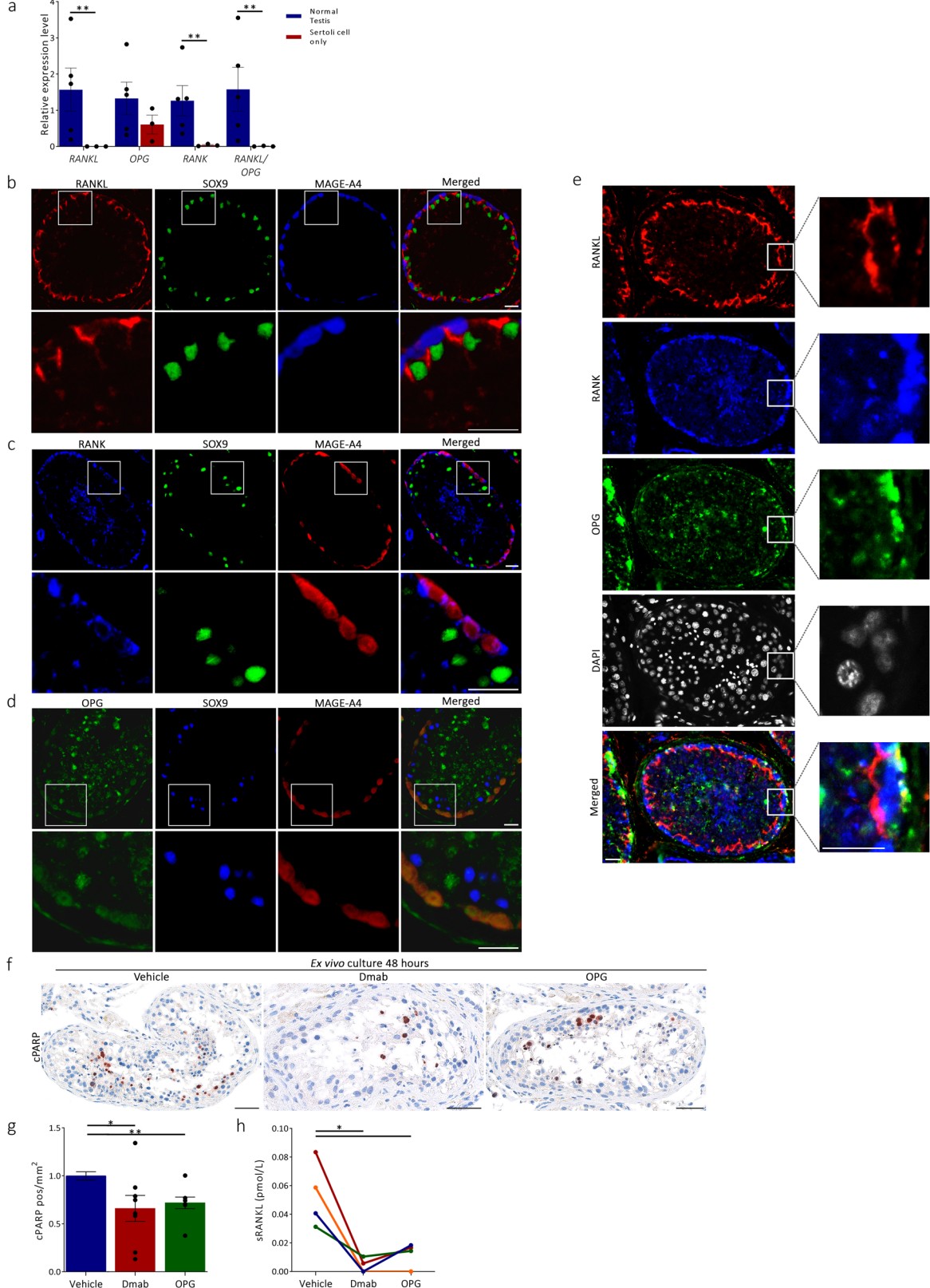

increased germ cell proliferation was observed (Supplementary Fig. 6b). The comparable effect of OPG and Denosumab on germ cell apoptosis suggests that RANKL inhibition exerts stronger cell cycle regulation than OPG-mediated TRAIL inhibition. Analysis of the media after 24–48 h of culture showed that sRANKL concentration declined over time (Supplementary Fig. 6c).

Interestingly, the release of sRANKL into the culture media was significantly reduced by Denosumab and OPG treatment compared with vehicle controls ($p < 0.05$) (Fig. 3h).

**RANKL signaling in male reproductive tract and human spermatozoa.** Interestingly, RANKL was abundantly expressed in

**Fig. 3 Expression of RANKL, RANK, and OPG in human testis and effects of RANKL inhibition. a** Gene expression of *RANKL*, *RANK*, and *OPG* in normal human testis (blue) and Sertoli cell only (red), ($p = 0.002$, $p = 0.003$, $p = 0.002$). **b** Triple immunofluorescence with RANKL (sc-7628, red), SOX9 (Sertoli cell marker, green), and MAGE-A4 (germ cell marker, blue) in normal testis. **c** Triple immunofluorescence RANK (HPA0277728, blue), SOX9 (green), and MAGE-A4 (red) in normal testis. **d** Triple immunofluorescence with OPG (sc-21038, green), SOX9 (blue), and MAGE-A4 (red) in normal testis. **e** Triple immunofluorescence with RANKL (sc-7628, red), RANK (HPA0277728, blue), OPG (sc-21038, geen), and DAPI (gray) in normal testis. **f** Apoptotic germ cells in ex vivo cultures treated with vehicle, Denosumab (100 ng/ml), or OPG (50 ng/ml) for 48 h investigated by IHC staining for cleaved PARP (cPARP, marker of apoptosis). Counterstaining with Mayer's hematoxylin. **g** Number of cPARP positive cells per area after 48 h of treatment with vehicle (blue), Denosumab (red), or OPG (green), ($p = 0.002$, $p = 0.041$). **h** Paired measurement of sRANKL in the media of ex vivo testis cultures (indicated by different colors) following 48 h of vehicle, Denosumab, or OPG treatment, ($p = 0.027$, $p = 0.047$). Scale bars correspond to 25 μm (**b**-**e**) or 50 μm (**f**). Data presented individually and as mean ± SEM, with n (normal testis/Sertoli cell only) = 5/3 (**a**), n (vehicle/Dmab/OPG) = 8/8/8 (**g**), and n (vehicle/Dmab/OPG) = 4/4/4 (**h**). Statistical test: unpaired two-sided Student's *t* test (**a**) or paired two-sided Student's *t* test (**g**, **h**) with *$p < 0.05$, **$p < 0.01$. Abbreviations: Dmab Denosumab.

the luminal cytoplasm of all human epididymal compartments, prostate gland, and seminal vesicle (Fig. 4a, b). The expression varied in the different organs but there was abundant expression in the luminal cytoplasm of the epithelial cells in corpus, cauda epididymis, and prostate. RANK was also expressed in caput and cauda epididymis, prostate, and seminal vesicle with the strongest expression in cauda epididymis (Fig. 4a, b). In contrast, OPG was expressed predominantly in the basal cells surrounding the luminal epithelial cells and in very few epithelial cells on the luminal site of epididymis, prostate, and seminal vesicle (Fig. 4a, b). RANKL was also expressed in the acrosomal region of most human spermatozoa (Fig. 4c), which suggests a possible role in mature spermatozoa. Denosumab induced no change in intracellular calcium concentration in human spermatozoa and had no relevant effects on the subsequent progesterone response, which is an important stimulating signal during fertilization (Fig. 4d). Moreover, Denosumab had no effect compared with vehicle treatment on the acrosome reaction (Fig. 4e).

**Soluble RANKL: high in seminal fluid, inversely associated with semen quality, and separates normal from infertile men.** Serum OPG and soluble RANKL (sRANKL) were inversely associated ($p < 0.005$) in 300 infertile men participating in Copenhagen Bone Gonadal study (NCT01304927) (Supplementary Fig. 7a; Supplementary Table 6). Interestingly, seminal fluid levels of sRANKL were approximately 100-fold higher than corresponding serum levels (average serum sRANKL = 0.27 pmol/L vs. seminal 26.19 pmol/L) in both infertile and normal men (Fig. 5a). There was no correlation between seminal and serum sRANKL concentration. Instead, Infertile men had higher seminal sRANKL concentration ($p < 0.0005$) and lower serum sRANKL levels ($p < 0.0005$) than healthy men (Fig. 5a). Seminal fluid sRANKL concentrations were significantly negatively associated with all semen quality variables except for semen volume (Fig. 5c; Supplementary Fig. 7b). Seminal fluid sRANKL remained inversely associated with total number of sperm (β-0.010, $p = 0.043$), sperm concentration (β-0.012, $p = 0.015$), progressive motility (β-0.210, $p = 0.001$), sperm motility (β-0.132, $p = 0.0120$), sperm morphology (β-0.026, $p = 0.025$), total number of progressive motile sperm (β-0.018, $p = 0.001$), and total number of morphologically normal sperm (β-0.019, $p = 0.002$) after adjustment for duration of abstinence (Fig. 5c; Supplementary Fig. 7c). All associations remained statistically significant after adjustment for duration of abstinence and BMI except for total sperm count and morphology (Supplementary Table 7). The seminal/serum sRANKL ratio was also negatively associated with all semen quality variables even after adjustment for relevant confounders. Seminal sRANKL concentration and seminal/serum sRANKL ratio were significantly higher in poor versus good semen quality groups after stratification according to WHO criteria and adjusted for duration of abstinence; WHO threshold for total number of

sperm 40 million/sample ($p = 0.056$), sperm concentration = 15 million/mL (NS), severe oligospermia sperm concentration <5 million/mL (mean 3.1 vs. 2.9, $p = 0.035$), low motility <40% (mean 3.1 vs. 2.8, $p = 0.002$), low sperm morphology <4% (mean 3.1 vs. 2.8, $p = 0.006$) (Fig. 5d, e). sRANKL in seminal fluid was comparable when assessed in 31 infertile men delivering two semen samples on average 14 days apart with a high correlation between the two samples (Spearman correlation $r = 0.91$) despite of a dilution (1:60) of samples prior to analyses (Fig. 5b; Supplementary Fig. 7d). Receiver operating characteristic (ROC) analysis was used to determine whether seminal and seminal/serum sRANKL had any predictive value for distinguishing infertile from normal men compared with semen quality variables (Fig. 5f). Seminal fluid RANKL had an area under the curve (AUC) of 67% [AUC = 0.67] thus exceeding the predictive value for total sperm count [AUC = 0.66], but with lower predictive value than sperm motility [AUC = 0.85] and morphology [AUC = 0.79]. Interestingly, seminal/serum RANKL ratio had an [AUC = 0.83] and was a better predictor than all other investigated variables except for sperm motility. Seminal sRANKL concentrations were positively associated with serum estradiol (β 0.007, $p < 0.0005$), and negatively with testosterone (β-0.30, $p = 0.001$) and testosterone/LH ratio after adjustment for BMI (Fig. 5g).

**Denosumab lowers RANKL levels in seminal fluid and increases serum Inhibin B in infertile men.** To test efficacy and safety of RANKL inhibition in infertile men, a single dose of Denosumab (Prolia®, 60 mg) was injected subcutaneously into 12 infertile men (Supplementary Table 8). Semen quality, reproductive hormones, calcium homeostasis, sRANKL, and OPG were evaluated at day 5, 20, 40, 80, 120, and 180 after injection to avoid missing an early anti-apoptotic effect (day 5, 20, 40), but most importantly to assess semen quality after a full cycle of spermatogenesis (74 days) on day 80. Therefore, the prespecified statistical analysis stated that direct comparisons between timepoints starting at day 80 with baseline should be conducted initially to avoid adjusting for multiple comparisons. Semen quality variables and hormones were normalized to the average of the two samples delivered prior to treatment start. One man was lost to follow up and one man had high fever at the beginning of the trial and was therefore per protocol excluded from the analysis leaving 10 men in the study for all semen quality studies. Serum levels of AMH and Inhibin B were significantly higher at day 80 ($p = 0.003$ and $p = 0.018$) compared with baseline (Fig. 6a; Supplementary Fig. 8f). Inhibin B levels increased in most men when comparing baseline with day 80 (Fig. 6d). Serum levels of FSH did not change significantly, but the Inhibin B/FSH ratio increased significantly at day 80 ($p = 0.02$) (Fig. 6a; Supplementary Fig. 8e). Serum estradiol concentration was significantly lower at day 180 ($p = 0.001$), while testosterone, SHBG, and LH concentrations remained unchanged (Supplementary Fig. 8a–d). Total sperm

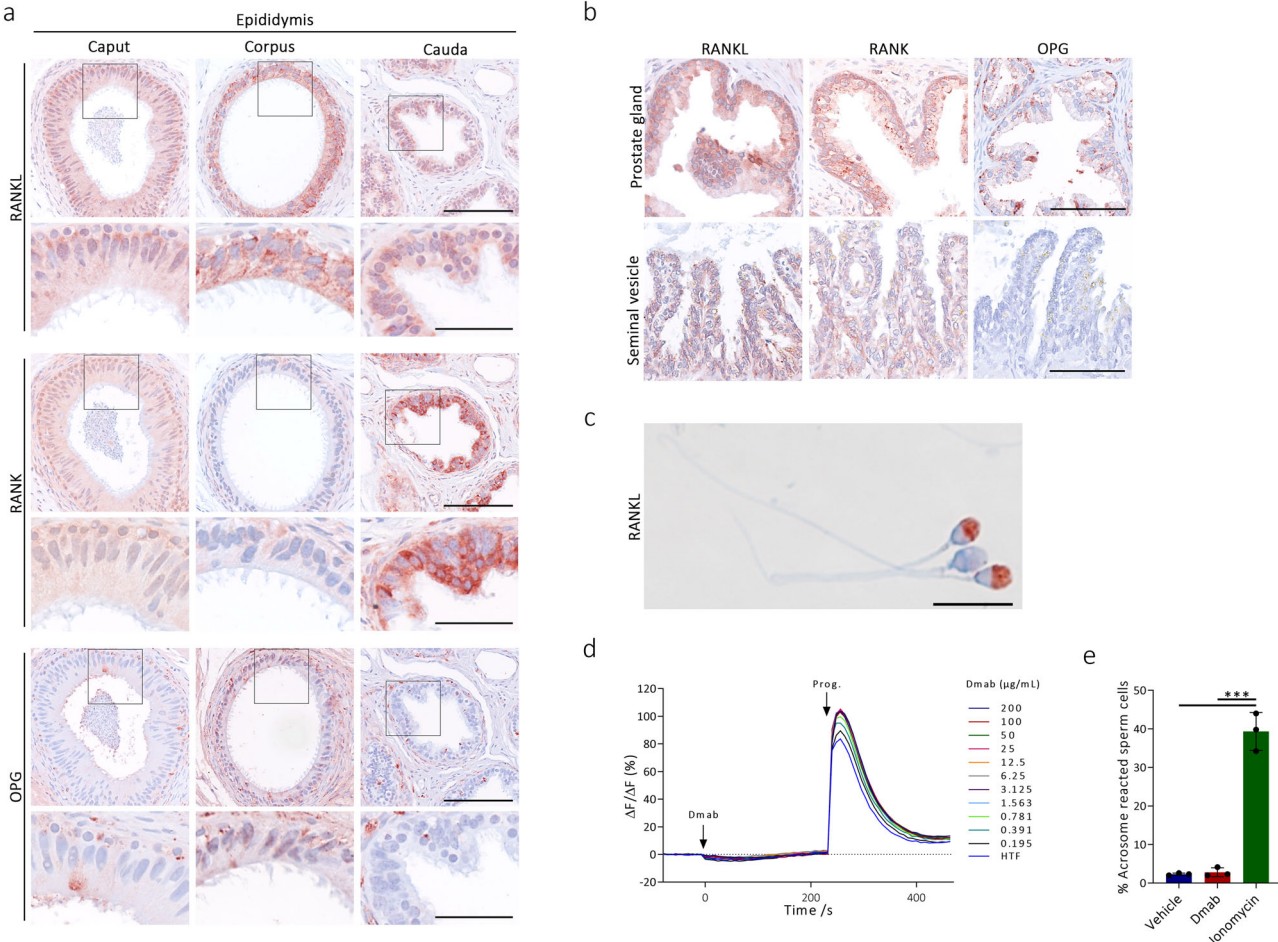

**Fig. 4 Expression of RANKL, RANK, and OPG in male reproductive tract and effects of denosumab on human spermatozoa. a** Expression of RANKL (sc-9073), RANK (sc-9073), and OPG (sc-8468) in human caput, corpus, and cauda epididymis. Counterstaining with Mayer's hematoxylin. Scale bars correspond to 125 µm for low magnification (upper panels) and 50 µm for high magnification (lower panels). Black inserts mark high magnification areas. **b** Expression of RANKL (sc-9073), RANK (sc-9073), and OPG (sc-8468) in human prostate gland and seminal vesicle. Counterstaining with Mayer's hematoxylin. Scale bars correspond to 100 µm. **c** Immunostaining of RANKL (sc-9073) in human spermatozoa. Counterstaining with Mayer's hematoxylin. Scale bar corresponds to 10 µm. **d** Intracellular $Ca^{2+}$ signals induced by Denosumab in concentrations from 200 µg/mL to 0.195 µg/mL (indicated by different colors). ΔF/F0 (%) indicates the percentage change in fluorescence that translates into cytoplasmic calcium changes. Progesterone (100 nM) was used afterwards to test for an augmented effect, while human tubular fluid (HTF) was used as a negative control (repeated twice in three donors). **e** Acrosome reacted spermatozoa (% of number of spermatozoa) from healthy donors after treatment with Denosumab (6 µg/mL, red), the positive control ionomycin (10 µM, green), or vehicle treatment (HTF with 0.2% DMSO, blue), ($p < 0.0001$, $p < 0.0001$). Data presented as mean (D) or mean ± SEM (**e**) with $n = 3$ (**d**, **e**). Statistical test: one-way analysis of variance (ANOVA) with Dunnett's test to adjust for multiple comparisons (**e**) with ***$p < 0.001$. Abbreviations: Dmab Denosumab, Prog. Progesterone.

count, sperm concentration, sperm motility, and total number of progressive motile sperm did not change at day 80 (Fig. 6b, c; Supplementary Fig. 9a). However, at day 80 the number of progressive motile spermatozoa ($p = 0.057$) and motile spermatozoa ($p = 0.071$) tended to be higher compared with baseline (Fig. 6e; Supplementary Fig. 9b). Interestingly, there seemed to be a subgroup of the men (60%) who experienced an increase in the number of progressive motile sperm (between 100–500%), while the remaining 40% experienced a marked decrease in number of progressive motile spermatozoa (Fig. 6e). This preliminary finding could indicate that the effect of Denosumab is dependent on specific baseline characteristics of the patients. Assessment of clinical and biochemical differences at baseline indicated that high serum OPG prior to treatment was the best predictor for a positive treatment response to Denosumab (Fig. 6f, g and Supplementary 9c). Men with OPG levels >3.0 pmol/L had a significant increase in total motile sperm and total progressive

motile sperm after Denosumab treatment at day 80 (both, $p < 0.05$) (Fig. 6g; Supplementary Fig. 9c). High baseline serum Inhibin B and AMH were also positive predictors (Fig. 6h, i). Importantly, these possible biomarkers are preliminary findings of biological interest, but cannot be considered clinically relevant before they have been tested and validated prospectively in placebo-controlled trials. We also tested whether Denosumab increased sperm DNA fragmentation index (DFI) at day 20 and 120, but no difference in DFI compared with baseline was found (Fig. 6j). Noteworthy, sRANKL concentration in serum was completely repressed instantly in all men following injection of Denosumab and was detectable again (except in one man) at day 120 or 180 (Fig. 6k). In contrast serum OPG increased slightly at day 20 and 40 (although not significantly) after denosumab injection before returning to baseline levels (Fig. 6k). Furthermore, Denosumab lowered sRANKL levels in the seminal fluid, but seminal sRANKL concentration was not completely repressed

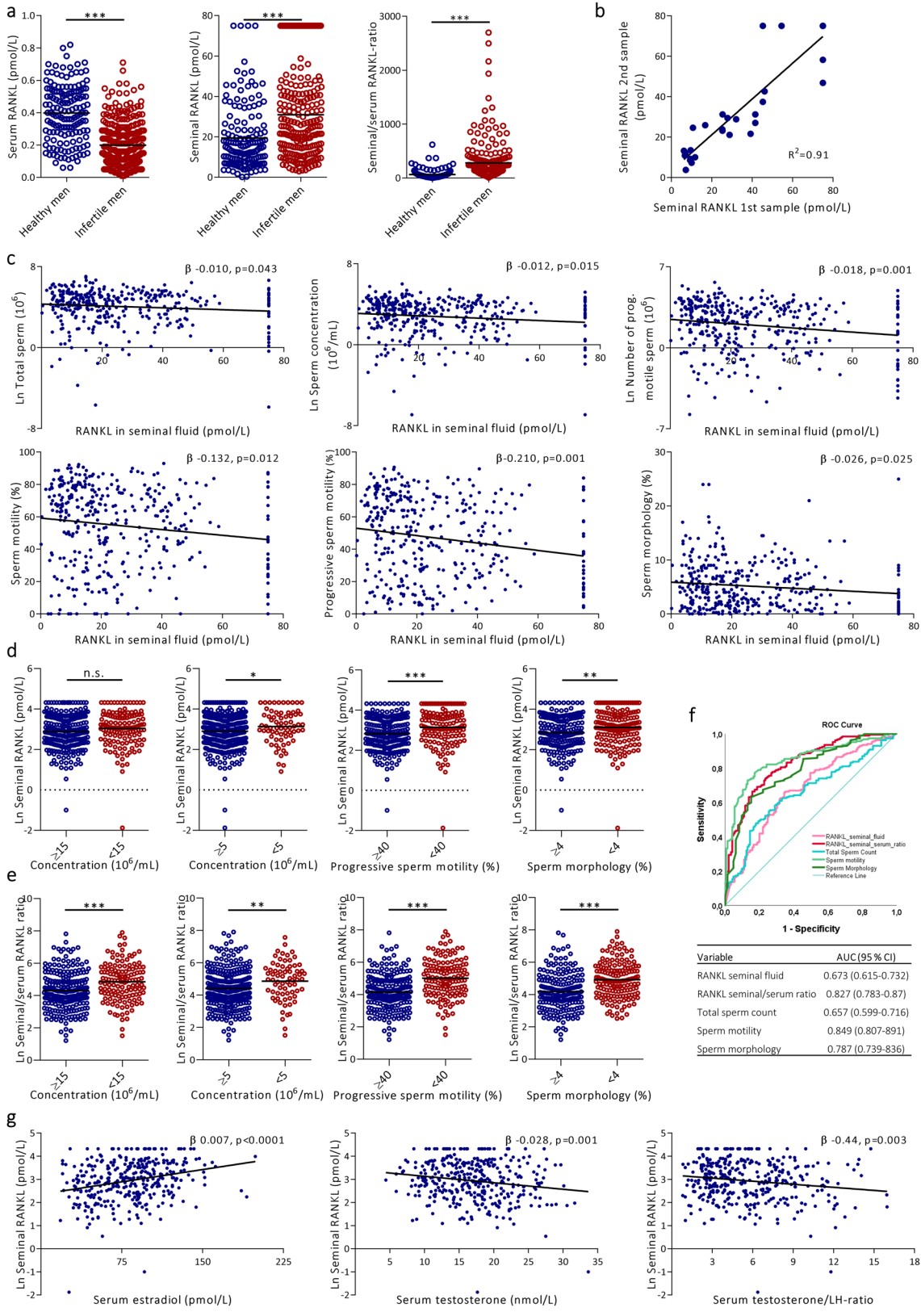

and remained detectable and higher than serum levels in most men at day 80 and 120 (Fig. 6l). Noteworthy, 2 of the men who completed the study were responsible for a pregnancy during the study and their spouses conceived naturally with two healthy live births.

## Discussion

The present study demonstrates that the RANKL-system is present in the male reproductive organs of both mice and humans. Presence of RANKL in Sertoli cells, spermatocytes, and spermatids, the receptor RANK in spermatogonia and spermatids, and

**Fig. 5 Serum and seminal fluid concentrations of sRANKL in relation to semen quality. a** (left) Serum sRANKL (pmol/L), (middle) seminal sRANKL (pmol/L), and (right) seminal/serum sRANKL ratio in healthy (blue) and infertile men (red), (left panel; $p < 0.0001$, middle; $p < 0.0001$, right; $p < 0.0001$). **b** Longitudinal measurements of seminal sRANKL levels in 31 infertile men with an average of 14 days between sample collections. **c** Semen quality variables and seminal sRANKL levels in a pooled linear regression model of both healthy and infertile men. **d** Seminal sRANKL levels and semen quality stratified in groups according to WHO references for normal (blue) vs. low (red) semen quality, (left panel; $p = 0.096$, left middle; $p = 0.041$, right middle; $p = 0.0009$, right; $p = 0.0034$). Pooled analyses of all healthy and infertile men. **e** Seminal/serum sRANKL ratios and semen quality stratified in groups according to WHO classification of normal vs. low semen quality, (left panel; $p < 0.0001$, left middle; $p = 0.005$, right middle; $p < 0.0001$, right; $p < 0.0001$). Pooled analyses of all healthy and infertile men. **f** Receiver operating characteristic (ROC) curve analysis and table showing sensitivity and 1-specificity for semen quality variables, seminal sRANKL, and sRANKL seminal/serum ratio as well as area under the curve (AUC) with 95% confidence intervals (CI). **g** Sex steroid levels in serum and seminal sRANKL levels in a linear regression model of both healthy and infertile men. All beta and $p$-values are adjusted for duration of abstinence. Statistical tests: two-sided Student's $t$ test (**a**, **d**, **e**) and linear regression model (**c**, **g**) with *$p < 0.05$, **$p < 0.01$, ***$p < 0.001$.

the inhibitor OPG in spermatogonia and peritubular cells highlights the dependence of intimate cell–cell interaction in the gonad as has been shown for RANKL signaling in bone[19,28]. A reproductive effect was shown in mice with Sertoli cell-specific suppression of RANKL that presented with increased testicular weight and sperm production thereby demonstrating a regulatory role of RANKL in Sertoli-germ cell interaction and spermatogenesis. Mice with global deficiency of RANKL presented with increased male fertility but the exact mechanism of action was not demonstrated as the increase in sperm production and sexual activity was not statistically significantly different from control mice. However, the proposed stimulatory effect of global RANKL inhibition on gonadal function was supported by the increase in testicular weight and sperm count in WT mice treated with OPG for 2 weeks. OPG was used because Denosumab only blocks RANKL in higher primates and cannot be examined in rodent models[29]. Denosumab is an OPG-like antibody that binds all isoforms of RANKL but not to other members of the TNF-family including TRAIL[29,30], while OPG binds both RANKL and TRAIL[31]. The rapid increase in testicular weight and sperm output following OPG treatment was surprising because spermatogenesis in mice takes 35 days. However, prolonged treatment with OPG for five weeks had no effect on testicular weight or sperm counts. The lack of effect after five weeks of treatment may be explained by autoantibodies, exhaustion, or compensation and is in line with long-term Denosumab toxicology studies in monkeys[32,36], while the observed increase in sperm production after two weeks of treatment could be due to decreased apoptosis rather than increased germ cell proliferation. TRAIL, a potent apoptosis-inducer in the testis is effectively blocked by OPG[33,34], and TRAIL inhibition could in theory lead to less apoptosis and thus a faster increase in sperm counts[33,35]. Interestingly, the observed compensatory low testicular OPG expression in mice with global RANKL deficiency may facilitate increased TRAIL-mediated apoptosis. This could explain why there was no change in testicular weight compared with control mice while the increase in testicular weight in Sertoli RANKL-deficient mice may be due to lack of compensatory change in testicular OPG and TRAIL signaling. Injection of OPG into global RANKL-deficient mice suppressed fertility despite of normal sexual activity, which indicates that the effect of RANKL inhibition on fertility may depend on the presence of testicular RANKL and OPG activity.

Importantly, the testicular expression pattern of RANKL signaling is overall conserved between mice and humans. Ex vivo cultures of human testicular tissue allowed us to compare the effects of OPG and Denosumab treatment, used in concentrations lower than actually measured in seminal fluid following subcutaneous injection[36], on human germ cell apoptosis and proliferation. The comparable anti-apoptotic effects on human germ cells and suppression of RANKL released into the media of OPG and Denosumab treatment show that the effect is mediated through RANKL signaling rather than by TRAIL. Denosumab

and OPG did not significantly increase the number of proliferating germ cells in ex vivo cultures, which suggests that the effect on sperm counts most likely is the result of reduced germ cell apoptosis. Noteworthy, RANKL signaling may be dependent on the etiology and severity of the male infertility because OPG was markedly expressed in the Sertoli cells in Sertoli-cell-only tubules and was thus, clearly different from the peritubular/germ cell expression found in normal seminiferous tubules. Future studies are needed to investigate RANKL signaling during development and in dysgenetic testes to determine whether the change in expression is the result of impaired Sertoli cell maturation or secondary to the germ cell loss.

The marked expression of RANKL in the epithelia lining epididymis, prostate gland, and seminal vesicle indicates that inhibition of extra-gonadal RANKL may influence reproductive function. RANK was also expressed in all organs lining the reproductive tract, while OPG was found in few of the luminal epithelial cells of all organs lining the male reproductive tract. RANKL signaling may have a role in the epithelia but could also signal to the spermatozoa as expression of RANK and RANKL in spermatids and the acrosomal region of human spermatozoa supports a role during sperm maturation and maybe even during ejaculation. However, Denosumab induced no change in intracellular calcium concentration, the progesterone response, or the acrosome reaction in human spermatozoa. This indicates that the possible role of RANKL in human spermatozoa may be different and could in theory be involved in binding to the epithelia of the female reproductive tract or the cumulus cells surrounding the oocyte[37]. The 100-fold higher sRANKL concentration in seminal fluid compared with serum supports a biological role. It is plausible that RANKL is secreted, cleaved, or released from the epithelial cells in the reproductive tract under the control of local regulators as there was no link between serum and seminal concentrations of sRANKL. A biological role of seminal RANKL was indicated by the negative associations between seminal sRANKL concentrations and semen quality variables including the total number of progressive motile sperm and total number of morphological normal sperm that refers to the number of mature and functional sperm and not just quantity. The strong associations between seminal RANKL levels and serum testosterone and estradiol suggest that RANKL activity and release into the epididymis and prostate may be influenced by sex steroids. The associations between semen quality variables and seminal RANKL levels were not strong, which implies that other regulators such as FSH, estradiol, and testosterone likely exert a stronger regulatory effect on spermatogenesis and sperm maturation than seminal RANKL concentration.

Stratification of men according to WHO references for normal semen quality revealed that men with poor semen quality, including low number of sperm, low motility, or morphologically abnormal sperm had higher seminal sRANKL levels than men with normal semen quality[38]. Interestingly, infertile men had 50%

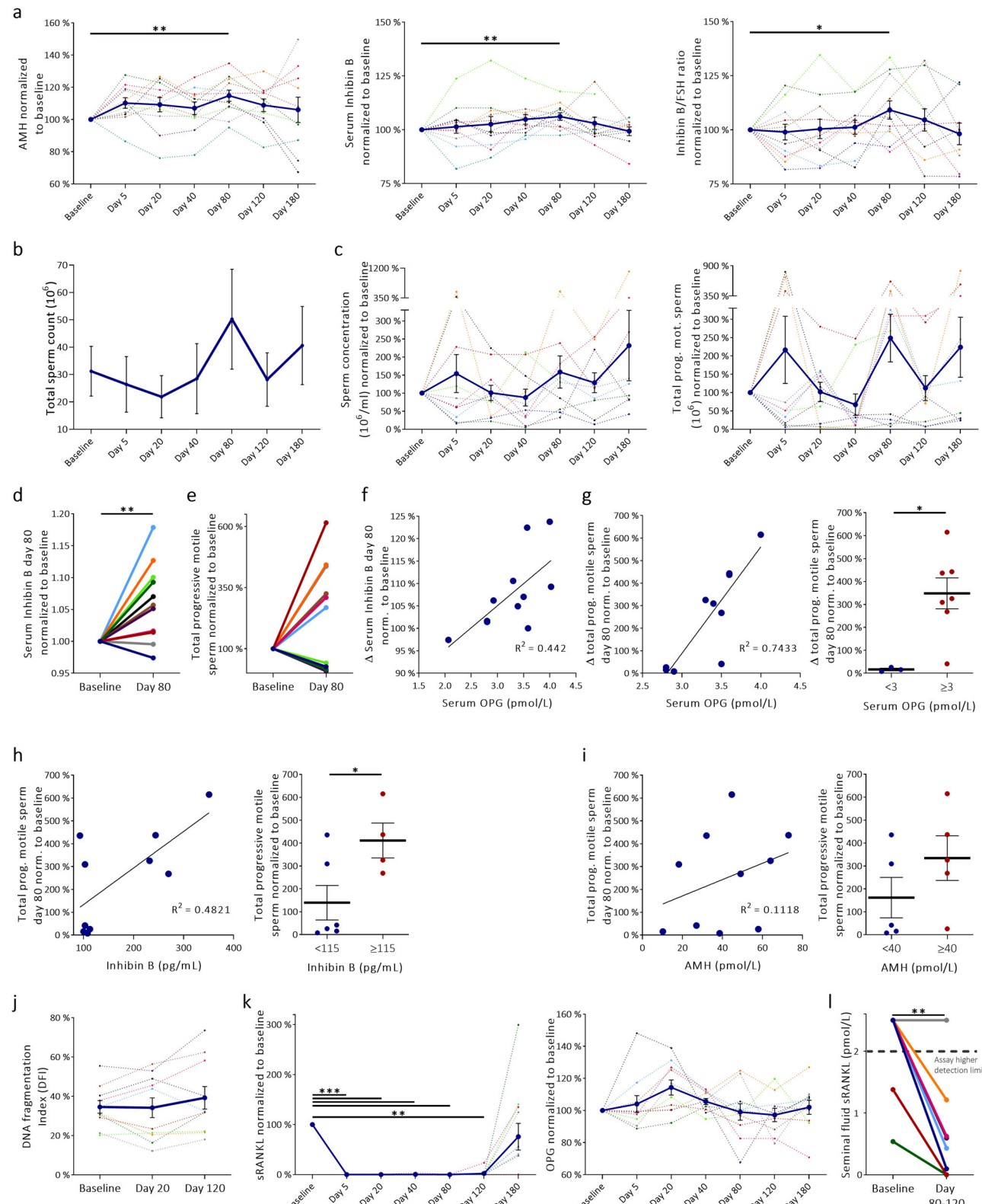

higher seminal sRANKL levels compared with normal men. In contrast, serum sRANKL levels were significantly higher in normal men. The ratio of seminal/serum sRANKL was therefore highly different between normal and infertile men and inversely associated with all semen quality variables. This suggests that the seminal/serum sRANKL ratio may be used as a marker to distinguish infertile from normal men although a considerable overlap exists. ROC curves showed that the AUC for seminal RANKL levels was higher than that for total sperm count and thus a better marker to identify infertile men. The seminal/serum RANKL ratio had an even higher AUC and was a better marker than all semen quality variables except for sperm motility. RANKL deficiency and conversely RANKL injection changed epididymal weight in mice, but we provide no evidence for a

**Fig. 6 Reproductive hormones and semen quality before and after treatment with Denosumab (60 mg) in 10 infertile men. a** Serum AMH (pmol/L), Inhibin B (pg/ml), and Inhibin B/FSH ratio, (left panel; $p = 0.004$, middle; $p = 0.001$, right; $p = 0.03$). **b** Total sperm count ($10^6$). **c** Sperm concentration ($10^6$/ml) and total number of progressive motile sperm ($10^6$). **d** Individual changes in serum inhibin B at day 80, ($p = 0.001$). **e** Individual changes in total progressive motile sperm at day 80. **f** Baseline serum OPG and treatment response evaluated by percental change in serum Inhibin B at day 80. **g** Baseline serum OPG and treatment response evaluated by percental change in total progressive motile sperm at day 80 normalized to baseline, (right panel; $p = 0.015$). **h** Baseline serum Inhibin B and treatment response evaluated by relative change in total progressive motile sperm at day 80 normalized to baseline, (right panel; $p = 0.042$). **i** Baseline serum AMH and treatment response evaluated by relative change in total progressive motile sperm at day 80 normalized to baseline. **j** DNA fragmentation at baseline, day 20, and 120. **k** Serum sRANKL and OPG normalized to baseline values, (left to right; $p < 0.0005$, $p < 0.0005$, $p < 0.0005$, $p < 0.0005$, $p = 0.002$). **l** Seminal fluid levels of sRANKL (pmol/L) at day 80 to 120 compared with baseline, ($p = 0.002$). Dotted line indicates upper reference limit for the ELISA. All hormones and semen quality variables are presented after normalization to mean of the two samples delivered prior to treatment start except in Fig. B. Note, y-axes and x-axes do not intersect at 0 (**a**, **b**, **d**-**f**, **k**). Data presented as mean ± SEM (**b**), individually (**d**-**l**), or combined mean ± SEM (blue) and individually (dotted) (**a**, **c**, **g**, **h**, **i**, **j**, **k**) with $n = 10$ (**a**-**k**), $n = 8$ (**l**) Statistical test: paired two-sided Student's $t$ test (**a**, **d**, **k**, **l**) and unpaired two-sided Student's $t$ test (**g**, **h**). *$p < 0.05$, **$p < 0.01$, ***$p < 0.001$.

direct negative effect of soluble RANKL available in the seminal fluid. Most IgG antibodies including Denosumab may have a restricted penetrance into the luminal site of testis or epididymis due to the blood–testis and blood–epididymis barrier[39]. However, pharmacokinetic studies conducted by Amgen have shown radioactive labeled Denosumab within the seminiferous tubules in higher primates after subcutaneous injections[40]. Furthermore, Denosumab is also detectable in the seminal fluid, which highlights its entry into the male reproductive system where it may mediate direct effects on Sertoli–germ interaction, spermatocytes, spermatids, spermatozoa, and on RANKL in the epididymis, prostate, or seminal vesicles[20].

For safety reasons we initiated a small intervention trial using a standard osteoporosis dose of Denosumab (60 mg, once) in infertile men to test gonadal effects. Denosumab caused an immediate suppression of serum sRANKL while serum OPG remained high. Interestingly, sRANKL concentration in seminal fluid was reduced but not fully repressed following treatment with the standard Denosumab regime for osteoporosis. It is unknown whether a more potent repression of seminal fluid sRANKL concentration may lead to an increased treatment response since it is likely that the optimal doses required for osteoporosis and improvement of male reproductive function differ. Analysis of multiple timepoints were included in this trial to avoid missing an early anti-apoptotic effect, but the prespecified statistical analysis states to initially compare endpoints after one length of spermatogenesis (74 days) with baseline to avoid adjustment for multiple comparisons. Denosumab induced an increase in serum AMH and Inhibin B after 80 days but had no significant effect on semen quality. Subgroup analysis showed that Denosumab increased the number of total progressive motile sperm in infertile men with high serum OPG levels, which supports a regulatory role of RANKL in male reproduction. However, this data should be interpreted with caution due to the non-placebo-controlled intervention and the small cohort size and should be considered preliminary. Serum Inhibin B and AMH were also candidates as predictive markers for a beneficial response, which implies that men with poor gonadal function or low RANKL activity may have a detrimental response as supported by the reduced fertility following OPG treatment in RANKL-deficient mice. The present study does not provide clinically valid information but biological insight by showing that Denosumab can lower seminal RANKL levels and improve reproductive function demonstrated by serum inhibin B and AMH and sperm output in men with the best gonadal function that corroborates the biological phenotypes shown in mice and the human ex vivo model. Moreover, the observed effects of OPG and Denosumab in our models indicate that reverse RANKL signaling is unlikely to be responsible for the observed effect on sperm production[25].

Although, it cannot be excluded that an unidentified endocrine or paracrine factor may suppress testicular OPG and augment the reproductive effects beyond inhibition of the Sertoli-germ cell interaction.

Seminal fluid sRANKL concentrations were positively associated with serum estradiol and negatively with serum testosterone. This observation is of great interest because infertile men often have lower testosterone/estradiol ratio than normal men[5,41] which may facilitate the high seminal sRANKL levels and would be in accordance with the regulatory role of sex steroids on RANKL and OPG expression in the skeleton. Noteworthy, Denosumab treatment in infertile men had no effect on serum gonadotropin or testosterone concentrations but induced a late and modest decrease in serum estradiol. Furthermore, almost all infertile men had an increase in AMH and Inhibin B levels, but only 60% of them were able to convert these endocrine changes into an increased number of progressive motile sperm. We cannot explain this, but it could be due to heterogenicity in the etiology of their infertility, which was supported by the finding that high serum levels of OPG, AMH, or Inhibin B were linked to a beneficial response. Further studies are required to elucidate the suggested dependence of RANKL activity and gonadal function for a beneficial response to RANKL inhibition in both animals and humans. The presented data should be considered a pilot study and prospective placebo-controlled trials are needed to investigate whether predictive biomarkers such as serum OPG levels, high seminal RANKL, or high Inhibin B or AMH can assist in the selection of the infertile men who may benefit from Denosumab treatment.

Safety is a concern when applying new drugs related to reproductive function and Denosumab needs to be carefully evaluated because it is a known teratogen. After the initial approval of the drug FDA requested measurements of Denosumab levels in the seminal fluid and these studies showed that transfer to the woman or the fetus was not likely to take place[36]. Moreover, subcutaneous Denosumab (Prolia®) is used clinically worldwide to treat osteoporosis and considered a safe treatment option with frequent mild and a few rare serious side-effects such as osteonecrosis of the jaw and atypical femoral fractures[42]. Another potential side-effect of RANKL inhibition could be impaired gamete quality due to increased DNA fragmentation as a result of reduced germ cell apoptosis. This issue was addressed by showing increased fertility in mice treated with OPG or with genetically repressed RANKL, while Denosumab treatment in infertile men induced no change in DNA sperm fragmentation and two healthy babies were born after treatment. Placebo-controlled randomized studies are required to clarify efficacy and safety before RANKL inhibitors can be considered as a clinical application for some cases of male infertility. Another safety concern could be growth of TGCTs that normally appear between

15–40 years of age and are more frequent among infertile men. The precursor for TGCT is known as GCNIS[14,43], and Denosumab treatment may in theory facilitate proliferation of premalignant germ cells, promote malignant transformation, or stimulate growth of an existing TGCT.

In conclusion, this translational study suggests that RANKL signaling is a regulator of male reproductive function by being a mediator in the Sertoli–germ cell interaction and in the male reproductive tract thereby exerting an influence on semen quality and male fertility.

## Methods

**Animal models**. Rankl floxed mice (*Rankl^fl/fl*), C57BL/6 (WT), and VasaCre mice were purchased from The Jackson Laboratory, and MisCre (*Amh:Cre*) mice were kindly provided by Jorma Toppari (Turku, Finland). Detailed breeding information can be found in Supplementary Methods 1. All mice were genotyped at weaning (P21) from tail biopsies by PCR. All animal studies were approved by the Danish Animal Experiments Inspectorate (license number 2011/561-2006). Animals were housed and cared for according to National ethical guidelines. Serum was collected by cheek pouch to perform analysis; macroscopic analyses includes body, testis, epididymal, long bone, kidney weight, and survival. Sex steroids were measured with LC-MS as described previously[10]. The number of animals used for each endpoint was determined based on the initial OPG treatment study in wildtype mice.

**Fertility tests and reproductive analysis of mouse models**. All the fertility studies were conducted blinded to genotype at Timeline Bioresearch (Sweden). Male mice with Sertoli specific or global Rankl deficiency were compared directly with *Rankl^fl/fl* littermates and one male was caged with three virgin female mice for 5 days. Subsequently, *Rankl^fl/fl*, Sertoli cell-specific (*AMHCre;Rankl^fl/fl*) or global RANKL-deficient mice (*VasaCre;Rankl^fl/fl*) (*n* = 3–6) were either treated with OPG-FC (1 mg/kg) or vehicle two times weekly for two weeks before being caged with three 10 week-old WT C57BL/6 female mice in one cage for 5 days. Plugs were assessed daily and after 5 days the males were removed. Number of litters and pups per litter were systematically recorded. Epididymal sperm count was performed by extracting sperm surgically from the caudal epididymis and ductus deferens of adult (16–17 weeks) male mice to determine sperm count. To retrieve sperm, cauda epididymis was excised and separated from adipose tissue and vessels and placed in 1 ml of Quinn's sperm washing medium (Origio). A cut in the mid to distal region of the cauda epididymis was made to mince the distal epididymis and to allow the sperm to be separated from the tissue. Material was incubated in 5% $CO_2$, 37 °C for 30 min and epididymis was subsequently removed. Sperm was mixed in the fluid and 10 µl was taken to determine the percentage of motile sperm by counting 200 cells in a phase contrast microscope. A motile sperm is defined as a cell with a head that moves the tail. The sperm suspension was deposited in a counting chamber and counted twice by a technician blinded to genotype to determine total sperm count. Sperm counts were also determined automatically by using NC-3000 (Chemometec) for comparison (Supplementary Fig. 10). Multiple transverse sections of seminiferous tubules from at least four different animals per genotype were randomly selected to evaluate testicular histology. H&E staining on paraffin was done to measure diameter of seminiferous tubules and height of the seminiferous epithelium. Diameters were only determined in round seminiferous tubule sections in stage VI–VIII. Morphometric analysis was performed following scanning of sections on a NanoZoomer 2.0 HT (Hamamatsu Photonics) and images captured using the software NDPview version 2.6.13 (Hamamatsu Photonics). Each testis was divided into fragments either snap-frozen or stored at −80 °C for RNA or protein extraction or fixed overnight at 4 °C in formalin or Bouin's solution and embedded in paraffin. Tissue preparation, RNA, cDNA preparation, and subsequent qRT-PCR, as well as Western blot were performed as described detailed in Supplementary Methods 3. For presentation of full scan blots see the Source Data file. Representative bands from each primer combination were sequenced for verification (Eurofins MWG GmbH, Germany) and primer sequences are listed in Supplementary Table 2. Changes in gene expression were determined by comparing relative gene expression to β2-microglobulin (Supplementary Methods 3).

*Copenhagen bone gonadal study NCT01304927*. A double blind randomized clinical trial in which 307 infertile men were included and subsequently supplemented with either cholecalciferol 300.000 IU daily followed by 1400 IU + 500 mg calcium daily or placebo for 5 months. The study was approved by the regional ethics committee and EMA approval no. 2010-024588-42, H-4-2010-138, and 2010124801. All men initially delivered two semen samples, underwent physical investigation including ultrasound of genitals, full body, columnar and hip DXA scan, delivered fasting serum samples prior to the intervention and only these data were used here. More details can be found in Supplementary Table 6 and at https://clinicaltrials.gov/ct2/show/NCT01304927?term=blomberg+jensen&rank=4. Reproductive hormones and sex steroids were measured as described previously[11].

**Human tissue samples and sperm function tests**. Patients were recruited from Department of Growth and Reproduction, Rigshospitalet, Denmark in accordance with the Helsinki Declaration after approval from the local ethics committee (H-17004362). Adult testis tissue was obtained from orchidectomy specimens performed due to testicular cancer and specimens with invasive cancer were discarded after evaluation of IHC staining with PLAP, OCT4, or D2-40 antibodies. The non-malignant tissue was used for these studies as described in detail in Supplementary Methods 4. Tissue fixation and preparation, RNA and cDNA preparation, as well as subsequent qRT-PCR and Western blot analysis were performed. qRT-PCR was performed using specific primers targeting each mRNA Primer sequences are included in Supplementary Table 3. Healthy sperm donors were used to study the effects on Denosumab on human spermatozoa. Intracellular calcium was determined in human motile sperm separated by 1 h swim-up before being loaded with an fluorescent $Ca^{2+}$ indicator Fluo-4 (10 µM) (Thermo Scientific, #F14201) and measured following Denosumab or progesterone treatment by using a fluorescence plate reader (Fluostar Omega, BMG Labtech). The acrosome reaction was analyzed following capacitation of motile sperm that were mixed with 5 µg/ml fluorescein isothiocyanate conjugated Pisum sativum agglutinin (Sigma-Aldrich, #L0770), 0.5 ug/ml propidium iodide (ChemoMetec, #910-3016), and 10 µg/ml Hoechst-33342 (ChemoMetec, #910-3015) with or without Denosumab, Ionomycin (10 µM) or 0.2% DMSO (control). Samples were mixed with 100 µL immobilizing solution containing 0.6 M $NaHCO_3$ and 0.37% formaldehyde and loaded in a A2 slide (ChemoMetec, #942-0001) and assessed in a NucleoCounter® NC-3000™ image cytometer.

**Ex vivo human testis cultures**. Human testis was cultured using a hanging drop culture approach described in detail (Supplementary Methods 4). The effects of 1 ug/ml RANKL, 100 ng/ml Denosumab, and 50 ng/ml OPG-FC was investigated after being added to culture media with 0.1% BSA for 24–72 h.

**Immunohistochemistry/immunofluorescence**. All tissues were stained immunohistochemically for RANKL, OPG, and RANK. Briefly, immunohistochemical (IHC) staining was performed according to a standard indirect peroxidase method as described in detail in Supplementary Methods 3. All experiments were performed with a negative control staining without the primary antibody. Serial sections and immunofluorescence double and triple staining were used to examine concomitant expression of RANKL, RANK, and OPG as described in detail in Supplementary Methods 3. A detailed description of the antibody dilutions, secondary antibodies, and retrieval buffers is found in Supplementary Table 4 and Supplementary Table 5.

**TNFSF11 inhibition and fertility: a prospective intervention study NCT02422108**. To address the relevance of RANKL inhibition on human male reproduction, Denosumab (Prolia) 60 mg was injected into 12 infertile men once and semen quality was monitored for 180 days (https://clinicaltrials.gov/ct2/show/NCT02422108). The study was approved by the local ethics committee (H-15001992) and was conducted and monitored according to GCP standard after informed consent was obtained from all participants. All 12 infertile men had impaired semen quality and the severity varied from mild to severe oligospermia (Supplementary Table 8). Their bone health evaluated by DXA showed normal BMD and all had normal serum calcium, phosphate, alkaline phosphatase, 25-OHD, and PTH levels at baseline. All men delivered 2 semen samples prior to treatment start and serum RANKL and OPG were measured concomitantly with reproductive and calciotropic hormones at baseline (Supplementary Table 5). In total 22 men were screened, and 12 men were included. All men received calcium and vitamin D supplementation prior to treatment. After treatment start all 12 men were requested to deliver semen and blood samples at day 5, 20, 40, 80, 120, and 180. The early time points were included to avoid missing a putative anti-apoptotic effect due to the fast response of RANKL inhibition observed in human ex vivo models and wild-type mice.

**Biochemical analysis**. Fasting serum samples and seminal fluid were analyzed at all timepoints using validated methodology as described in Supplementary Table 9. Fasting blood samples were collected between 8:00 and 10:00 AM. Serum was analyzed immediately for calcium (total and ionized), phosphate, and PTH. The remaining analyses were conducted on thawed serum samples.

**Semen analysis**. Semen samples were delivered in an adjacent room in the out-patient clinic and information on duration of abstinence, fever, and spillage was obtained. The two semen samples were delivered 10–16 days apart prior to treatment start and analysis was conducted exactly as described in detail previously[11,44]. Briefly, semen volume was determined by weighing, sperm concentration was determined using a Bürker-Türk hemocytometer, and total sperm count was calculated. Sperm morphology was assessed according to strict criteria on Papanicolaou-stained smears. Sperm motility classified as progressive motile (WHO class A + B), nonprogressive motile (class C), or immotile (class D) was determined in duplicate at two times and presented as AB or ABC motility. Spermatozoa DNA fragmentation was investigated at baseline and 20 and 120 days after intervention at SPZ Laboratory (Copenhagen, Denmark). In brief, 0.5 mL

semen was diluted with TNE buffer, then mixed and frozen directly in liquid nitrogen until fluorescent staining according to the sperm chromatin structure assay protocol and then analyzed using a FACSCalibur (BD Biosciences, San Jose, CA) flow cytometer. All samples were run blinded in duplicate and recording stopped after 5000 events.

**Statistics and reproducibility**. 5–18 male mice at 16–17 weeks of age were used from each genotype to reach statistical significance—this number was calculated (power calculation) based on previous experience. For micrographs (Figs. 1a–d, 2f, 3b–e, and 4a, b) tissues from at least three different mice (of each phenotype) or human specimens were analyzed. All analyses were done by technicians or researchers that were blinded to genotype and/or treatment. All values were expressed as mean ± SEM. $p < 0.05$ was considered statistically significant. The following phenotypic variables: sperm count, organ weight, motility, and gene-expression were tested using two-sided Student's $t$ test or one-way analysis of variance (ANOVA) with Dunnett's test to adjust for multiple comparisons. The same was done for human gene expression data. No outliers were excluded from analyses. Pregnancy rates in breeding studies were analyzed with Pearson's chi-squared test and ANOVA with Dunnett's test to adjust for multiple comparison for average pups per female and litter-size. Data were log-transformed when appropriate. Cross-sectional data from Copenhagen Bone Gonadal Study were expressed as mean ± SEM except for sperm concentration and total sperm count (median). Associations between serum or seminal fluid levels of RANKL and OPG with semen quality and reproductive hormones were conducted after natural logarithmic transformation and adjustment for BMI, age, and duration of abstinence for all semen variables. Subsequently, all men were stratified according to WHO criteria for semen quality variables. Evaluation of gaussian distribution was done by plotting residuals and as a result the following variables were transformed with natural logarithm: RANKL in seminal fluid, RANKL in serum, seminal/serum RANKL ratio, total number of sperm, semen concentration, concentration of progressive motile sperm, and concentration of morphological normal sperm. Seminal RANKL levels were set as dependent in linear regression analyses with hormones, whereas it was set independent in linear regression with seminal parameters. RANKL levels, hormonal levels, and seminal parameters in infertile men were compared with levels in in infertile men and healthy men followed by a pooled analysis of all the men. All hormonal analyses were adjusted for BMI and analyses on semen quality were also adjusted for duration of abstinence. IBM SPSS statistics version 25 was used for analysis.

**TNFSF11 inhibition and fertility**. The primary endpoint was changes in sperm production from day 80 and onwards compared with baseline. The following predefined secondary endpoints were also evaluated: sperm DFI, FSH, Inhibin B, AMH, serum OPG, and sRANKL. Based on the prespecified statistical analyses each timepoint was compared with baseline by paired $t$-test starting with day 80. Subjects who terminated participation after visit day 1 but before visit day 180 were included for data analysis. Men with fever above 38.5 °C were included in the analyses 3 months after the fever episode. Spermatogenesis takes 74 days in men and the initial investigation was to determine the difference to all timepoints and calculate average for days 80, 120, and 180 and compare with baseline values to determine the effect of RANKL inhibition after a full length of spermatogenesis. Subsequent analyses were conducted after stratification into predefined subgroups according to BMI, serum RANKL, and OPG before treatment.

**Reporting summary**. Further information on research design is available in the Nature Research Reporting Summary linked to this article.

## Data availability

Source Data are present with this paper and all relevant data are available from the authors Source data are provided with this paper.

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

## Acknowledgements

Special thanks to Niels Jørgensen and Lise Akseglæde for assisting with recruitment of infertile and normal men, Brian Vendelboe and Ana Ricci for expert technical assistance, Anders Rehfeld for intracellular calcium measurements in human sperm, Hanne Frederiksen for steroid analyses of mouse serum, and the nurses and secretaries involved with the clinical studies. We thank Geert Carmeliet and Lisbeth Lieben for sharing and collecting tissues from VDR KO mice, Henrik Leffers for scientific advice and for providing micro-dissected data. We greatly appreciate Dr Jinho Kang and Dr. Kyong Hwa Soul, Korea for sharing their RANKL antibody. We also thank Rigshospitalet, Novo-Nordisk Foundation, Aase and Ejnar Danielsens Foundation, Danish Cancer Society and Danish Agency for Science, Technology and Innovation, Bioinnovation institute, vækstfonden.

## Author contributions

M.B.J.: Conception, draft of manuscript, design of studies, data analysis. C.H.A.: rodent models, ex vivo culture, IHC/WB, data analysis and layout. A.J.Ø.: ex vivo culture and tissue handling. J.E.N.: IHC and IF. I.B., L.J.M., P.S., R.B., B.L., and A.J.U.: design of studies, analysis of data. All authors: revision and approval of the final manuscript.

## Competing interests

All authors declare no competing interests except for M.B.J. who holds two patents on the use of RANKL inhibitors to treat male infertility. R.B. has been on the advisory Board of Amgen and B.L. is now employed by Radius Health. All other authors state no conflicts of interest.
