## [Peer Review File · Nature Communications]

Reviewers' comments:

Reviewer #1 (Remarks to the Author):

Infertility is a common problem affecting up to 15% of reproductive-aged couples worldwide. Male infertility is often associated with impaired semen parameters, but no treatment exists so far to improve semen quality. The authors are introducing RANKL-signaling as a novel regulator of male reproductive function by mediating interactions between Sertoli-germ-peritubular cells within the testis and in the male reproductive tract, to influence semen quality and male fertility. The testicular expression pattern of RANKL-signaling is conserved between mice and humans. The role play by RANKL-signaling in male reproduction was evaluated in vivo using different mouse models and ex vivo culture of human testicular tissue. They found that infertile men had higher seminal sRANKL levels and lower sRANKL serum levels than healthy men suggesting that the seminal/serum sRANKL ratio may be used as a marker to distinguish infertile from normal men. Finally, they performed a small human trial showing an increase progressive motile sperm cells in a subgroup of 6 infertile men (out of 10) following Denosumab injection.

This is an original and novel translational study suggesting a conserved regulatory role for RANKL in male reproduction. This should have a major impact in the field of human reproduction both at the clinical and fundamental levels. Although I recognize the difficulty of working in parallel with mouse and human models, it seems to me that the credibility of such a study would be strengthened if certain results were validated by complementary but independent experiments or, to compensate for the heterogeneity of the results, by increasing the number of measurements. In addition, the mechanism by which Rankl signaling affects spermatogenesis/sperm physiology is only partially addressed.

Relevant issues

1) RANKL expression in the male genital tract and high concentration in human seminal fluid suggest a role during sperm maturation and possibly oocyte fertilization. It would be relevant to test either in mice or better in human, how OPG or denosumab treatment affect sperm physiology, in particular intracellular Ca, progesterone-dependant Ca response as well as acrosomal reaction.

2) the expression study of RANKL, RANK and OPG in the testis was performed by IF on sections of human or mouse testis. In order to confirm the expression of RANKL in Sertoli (SC) cells and RANK in spermatogonia, it seems important to perform a double IF with specific markers of SC and spermatogonia. Currently, the identity of cells is deduced only according to their position in the seminal epithelium.

3) the fertility test performed with mutant Rankl mice should be better described. First of all, in mice, the fertility rate is highly dependent on the genetic background (pups/litter is 5-6 in a C57B6 genetic background and approx 15 in a CD1 outbred strain). As mice have undergone multiple crosses to generate the relevant genotypes, it is very likely that the genetic background must be mixed and specific for each line (SC-Rankl and Rankl -/-). It is therefore imperative to compare control mice and mutant mice of the same litters in order to minimize the variation of the genetic background. As described in Figure 2H, the control mice are referred to as WT. If these mice are really WT, they are not the right controls. Such a study should, therefore, compare the mutant SC-Rankl with control mice from the same litters that generated the SC-Rankl mutant animals. The same procedure should apply for the fertility test of the Rankl-/- mice.

4) Often, the results are very heterogeneous with great variability in the values obtained. This requires increasing the number of analyses to obtain meaningful values and credible results. Too often, the number of n is limited. this is the case for the mouse work (e.g. results described in Figures 1B, C, 2I, 3D-E). On the same topic, the title mentions the increase of "progressive motile sperm in some infertile men". However, the sample size is extremely low (N=10) and the subgroup mentioned is n=6. This is an important affirmation that is not supported enough with the existing data.

Minor changes:

p.5: "Anti-müllerian" should be anti-müllerian

p.6, l. 122 "Rankl expression in the testis varied considerably in WT mice because 50% had heterozygous loss of Rankl, ...". This is not possible either the 2 Rankl alleles are WT, or some of the control animals contain a defloxed Rankl allele. I believe that there is a confusion between WT and control animals. This has to be clarified in the text.

p.6: It would be more relevant to name the SC-specific deletion of Rankl "SC-Rankl" and the constitutive deletion of Rankl -/- or something equivalent. Naming them "Sertoli" and "global" is misleading

p.7: please define the acronym GCNIS

Reviewer #2 (Remarks to the Author):

The manuscript from Jensen et al, investigated the role of the RANKL, RANK and OPG in male fertility control by using in vivo approaches on global and conditional knockouts mice as well as ex vivo testicular cultures from human. While the RANKL/RANK/OPG system is well known in bone resorption process, this group is the first to report its function in the control of male reproductive functions. Acknowledging that there is no current treatment to face male infertility issues, the potential use of Denosumab as a FDA approved drug could have a major impact on male fertility control.

Comments:

1- According to the results provided in this manuscript, RANKL and RANK are both detected in the epididymis, prostate and seminal vesicles. While the authors suggest a role for the RANK system in sperm maturation, i.e. acquisition of sperm motility and fertilizing ability, it is worth narrowing down the potential contribution of epididymal RANK system to male fertility control. For instance, since RANK is present in elongated spermatid (Fig.1) does it remain on epididymal sperm? Is RANKL found secreted in the epididymal fluid? Could exogenous RANKL/OPG ratio (e.g. following OPG treatment) modulate sperm motility and acrosome reaction? Since Denosumab treatment increases progressive sperm motility in a category of infertile patients (Fig. 5D), a complementary mechanistic insight would be needed to better stratify the dichotomized responses observed following patient treatment.

2- As stated in the introduction, OPG presents with off-targets and a short half-life (1.80). The authors should justify the use of OPG to treat mouse animals instead of Denosumab.

3- l.150 "might be the result of reduced apoptosis". The authors should include the quantification of apoptosis following OPG treatment by TUNEL assay (or equivalent approach) to strengthen what was observed ex vivo (Fig 3DE) and to corroborate this hypothesis.

4- l.96 "RANK...with more prominent expression in spermatogonia". There is a discrepancy between the results obtained by IF and IHC for RANK. For instance, while IF demonstrates a predominant expression in elongated spermatids (Fig 1A), IHC shows a strong/almost exclusive expression in spermatogonia (Sup Fig1). Since the antibodies used for these two technics do not recognize the same epitope (C-term vs. N-term), the authors should discuss if a cleavage of RANK could explain the detection of two forms of the protein. If this is the case an additional level of regulation of the RANK/RANKL system could be envisioned in the testis, as it has been observed in the intestine (Zhao et al, FEBS letter, 2013 <https://febs.onlinelibrary.wiley.com/doi/full/10.1016/j.febslet.2013.08.005>).

5- Fig 2. The authors should specify what they mean by WT? Are the global and Sertoli-cell specific knockouts compared to C57BL/6 mice? To make sure that no unspecific phenotype is associated with either the presence of the recombinase itself or the floxed alleles, comparative controls should include Vasa:Cre, Amh:Cre, and ranklfl/fl mice.

6-l.154 "OPG treatment of Sertoli-cell RANKL deficient mice did not increase fertility, while OPG

treatment in global RANKL-deficient mice suppressed fertility all together and caused no pregnancies and litter size were zero in all six females". Please rephrase and discuss this result.

7. Fig 4. Correlative studies are performed on a global cohort of "infertile patients" vs. healthy men. While aetiology of male infertility is not always explained, the authors should stratify their cohort (i.e. asthenozoospermic, azoospermic, oligozoospermic patients) to perform additional correlations. This would help further discriminate the association of RANKL with sperm production and/or post-testicular maturation processes.

Minor comments:

1- I.93. "RANKL, RANK and OPG were detected at transcriptional and protein level". While protein detection somehow implies a transcriptional expression, there is no gene expression analysis shown in the mouse. Please remove "transcriptional" from the sentence.

2- I.372. Please define TGCT.

3. I.106. Please define OPG-FC

Reviewer #3 (Remarks to the Author):

Male infertility is a common condition with limited treatment options. The authors present a highly novel set of experiments investigating the role of RANKL as a regulator of male infertility. The paper is well written. The results create an overall convincing story which would have broad interest to the scientific community. I have some comments which require attention:

1. Many of the results are highly variable, as is usual especially with sperm parameters. Please present as mean +/-SD rather than SEM.

2. It is important to present serum testosterone levels for the mouse KO models, since testicular steroidogenesis will inevitably affect spermatogenesis. Without this, it is difficult to interpret the results.

3. Fig 4. I think it is important to be clear that the observed correlations between semen RANKL and other parameters are weak. Please discuss and speculate why.

4. Fig 4. Please also include correlation between semen RANKL and serum testosterone.

5. Fig 5. Given the small number of participants, I think post-hoc analysis of subgroup responses is highly suspect. I take your point that high OPG MAY signify likely response to therapy, but this clearly requires prospective, dedicated confirmation. Please highlight in discussion. Furthermore, please be explicit in the abstract that there was no change in sperm parameters in the recruited cohort of men with infertility.

To the Editor of Nature communications.

We appreciate the thorough and constructive comments from the three reviewers and Editors, which have helped us to improve the manuscript substantially. Below please find the specific answers to the reviewer comments:

Reviewer #1 (Remarks to the Author):

Infertility is a common problem affecting up to 15% of reproductive-aged couples worldwide. Male infertility is often associated with impaired semen parameters, but no treatment exists so far to improve semen quality. The authors are introducing RANKL-signaling as a novel regulator of male reproductive function by mediating interactions between Sertoli-germ-peritubular cells within the testis and in the male reproductive tract, to influence semen quality and male fertility. The testicular expression pattern of RANKL-signaling is conserved between mice and humans. The role play by RANKL-signaling in male reproduction was evaluated in vivo using different mouse models and ex vivo culture of human testicular tissue. They found that infertile men had higher seminal sRANKL levels and lower sRANKL serum levels than healthy men suggesting that the seminal/serum sRANKL ratio may be used as a marker to distinguish infertile from normal men. Finally, they performed a small human trial showing an increase progressive motile sperm cells in a subgroup of 6 infertile men (out of 10) following Denosumab injection. This is an original and novel translational study suggesting a conserved regulatory role for RANKL in male reproduction. This should have a major impact in the field of human reproduction both at the clinical and fundamental levels. Although I recognize the difficulty of working in parallel with mouse and human models, it seems to me that the credibility of such a study would be strengthened if certain results were validated by complementary but independent experiments or, to compensate for the heterogeneity of the results, by increasing the number of measurements. In addition, the mechanism by which Rankl signaling affects spermatogenesis/sperm physiology is only partially addressed.

Relevant issues

1) RANKL expression in the male genital tract and high concentration in human seminal fluid suggest a role during sperm maturation and possibly oocyte fertilization. It would be relevant to test either in mice or better in human, how OPG or denosumab treatment affect sperm physiology, in particular intracellular Ca, progesterone-dependant Ca response as well as acrosomal reaction.

We agree with the reviewer that the high RANKL concentration in the seminal fluid is of huge interest although it is very hard to determine the fraction that originates from the testis, epididymis, seminal vesicle, accessory glands, and prostate, respectively. To investigate this, we have extended this part of the manuscript, and we are now able to show the spatial expression of RANKL, RANK, and OPG in the organs lining the male reproductive tract in both mice and humans. Moreover, we show immunohistochemical expression of RANKL in the acrosomal region of human spermatozoa. It has previously been demonstrated by Amgen that Denosumab enters the seminal fluid in humans and we therefore tested the effects of Denosumab on intracellular calcium

and the acrosome reaction, and reassuringly found no effect. We believe this is of great interest as it implies that entry of Denosumab into the male reproductive tract is unlikely to induce the acrosome reaction or induce premature activation of the sperm. These new data have now been incorporated into Figures 4 and 5 and reported in the results section and discussion. We believe this was an important addition to the paper.

2) the expression study of RANKL, RANK and OPG in the testis was performed by IF on sections of human or mouse testis. In order to confirm the expression of RANKL in Sertoli (SC) cells and RANK in spermatogonia, it seems important to perform a double IF with specific markers of SC and spermatogonia. Currently, the identity of cells is deduced only according to their position in the seminal epithelium.

We agree with the reviewer and have in accordance with the suggestion conducted double and triple IF with a Sertoli cell marker (SOX9) and a germ cell marker (VASA/DDX4 or MAGE-A4) to show the cellular expression of RANKL, RANK, and OPG in mouse and human testis. These new data are presented in Figures 1 and 3 and in the results section.

3) the fertility test performed with mutant Rankl mice should be better described. First of all, in mice, the fertility rate is highly dependent on the genetic background (pups/litter is 5-6 in a C57B6 genetic background and approx.. 15 in a CD1 outbred strain). As mice have undergone multiple crosses to generate the relevant genotypes, it is very likely that the genetic background must be mixed and specific for each line (SC-Rankl and Rankl -/-). It is therefore imperative to compare control mice and mutant mice of the same litters in order to minimize the variation of the genetic background. As described in Figure 2H, the control mice are referred to as WT. If these mice are really WT, they are not the right controls. Such a study should, therefore, compare the mutant SC-Rankl with control mice from the same litters that generated the SC-Rankl mutant animals. The same procedure should apply for the fertility test of the Rankl-/- mice.

We apologize for not being precise and specific in the first submission. The reviewer can rest assured that the mice are not wildtype mice but Rankl-floxed littermates. We have described the fertility study in greater detail in the revised version of the manuscript. It should be noted that we initially intended to use the VASA-cre mice to generate a germ cell specific Rankl deficient mice. However, the VASA-cre became global which is a common phenomenon, but we discovered this rather late in the process and had already at this timepoint generated double VASA-cre +, AMH-cre +, Rankl fl/fl mice. Once we discovered this, we separated the two phenotypes again and used VASA-cre positive mice as the global model, and AMH-cre mice as the Sertoli cell specific model as demonstrated by tomato expression globally and specifically in the testis and supported by primers supplied by Dr. O'Brien that demonstrated global Rankl deletion. The only advantage of this huge workload is that the methodology used secures that the background is identical between the two presented mouse models since we used Rankl-floxed littermates without cre as controls in the fertility study and other endpoints. Moreover, the fertility study was conducted blinded to the genotype by independent technicians. We have revised the figure and highlighted this in the material & methods and results sections in accordance with the reviewer's suggestion.

4) Often, the results are very heterogeneous with great variability in the values obtained. This requires increasing the number of analyses to obtain meaningful values and credible results. Too often, the number of n is limited. this is the case for the mouse work (e.g. results described in Figures 1B, C, 2I, 3D-E).

We accept the critique raised by the reviewer and have extended the analysis and increased the number of mice in our study as shown in Figure 1B. There was no effect of OPG on sperm count or testis weight after 5 weeks treatment and we have also tried to use OPG-Fc for 5 weeks which resulted in similar results. However, it is not possible to combine these data as the two treatments are not identical and were not conducted in mice of similar age. Based on all the data generated we find that extension of the data presented in Figure 1F (prior 1C) would not be helpful. Moreover, there was no effect of OPG treatment on semen quality in RANKL deficient mice and we consider that these studies are not essential for the manuscript, so we have replaced Figure 2I with testosterone measurements.

Concerning figures 3D and 3E: These figures are not from mouse testis but human testis *ex vivo* culture. We have included n=8 for human testis samples without any invasive testis cancer cells present, which is a relatively high number. We had to discard many tissue samples because they contained malignant germ cells in the interstitial compartment (microinvasive). It was not an easy task to include healthy testicular tissues from 8 different men and we have been unable to increase sample size during the last six months even though we attempted this.

On the same topic, the title mentions the increase of “progressive motile sperm in some infertile men”. However, the sample size is extremely low (N=10) and the subgroup mentioned is n=6. This is an important affirmation that is not supported enough with the existing data.

We agree with the reviewer and have softened and included a more cautious statement in the abstract, revised the title/heading and the discussion accordingly.

Minor changes:

p.5: “Anti-müllerian” should be anti-müllerian

Corrected

p.6, l. 122 “Rankl expression in the testis varied considerably in WT mice because 50% had heterozygous loss of Rankl, ...”. This is not possible either the 2 Rankl alleles are WT, or some of the control animals contain a defloxed Rankl allele. I believe that there is a confusion between WT and control animals. This has to be clarified in the text.

We do understand that this statement may seem confusing, but this is a certain fact for all germ cell specific cre models. The phenomenon is greatly underreported in the literature. When you cross *Rankl^{fl/fl}* with *VasaCre-Tg* mice (*Cat# B6.FVB-Tg(Ddx4-cre)1Dcas/KnwJ*) or any germ cell specific cre line you will obtain either cre + or cre – mice with RANKL fl/wt. The VASA-cre positive pups will activate cre in the germ cells and delete RANKL on one of the alleles comprising the floxed Rankl. The germ cell will then undergo meiosis and form the haploid sperm harboring either Rankl wt or deleted Rankl as Rankl floxed was removed by the cre activity in the germ cell. Breeding this mouse with Rankl fl/fl will lead to either Rankl wt/fl or del/fl. Depending on the presence of VASA-cre the floxed RANKL will be excised in the germ cells so these mice will either be germ cell specific KO,

heterozygous or Rankl fl/wt. This highlights that Vasa-cre and all other germ cell specific cre lines only can generate germ cell specific loss of specific genes in combination with a heterozygous global deletion. The only way to circumvent this is by using an inducible cre such as Tamoxifen Vasa-cre which creates other problems. Fortunately, we were able to demonstrate this by using the primer combination that could show global deletion provided to us by Charles O'Brien (who generated the floxed Rankl mouse) and the use of td-Tomato as a marker for global/specific deletion in all mice.

p.6: It would be more relevant to name the SC-specific deletion of Rankl "SC-Rankl" and the constitutive deletion of Rankl -/- or something equivalent. Naming them "Sertoli" and "global" is misleading

We agree with the reviewer and have changed it to SC-RANKL and RANKL -/-.

p.7: please define the acronym GCNIS

We have now defined it the first time p. 7 l. 159 : "... dysgenetic samples with varying degrees of Sertoli-cell-only, spermatogenic arrest, and tubules containing germ cell neoplasia in situ (GCNIS)" .

Reviewer #2 (Remarks to the Author):

The manuscript from Jensen et al, investigated the role of the RANKL, RANK and OPG in male fertility control by using in vivo approaches on global and conditional knockouts mice as well as ex vivo testicular cultures from human. While the RANKL/RANK/OPG system is well known in bone resorption process, this group is the first to report its function in the control of male reproductive functions. Acknowledging that there is no current treatment to face male infertility issues, the potential use of Denosumab as a FDA approved drug could have a major impact on male fertility control.

Comments:

1- According to the results provided in this manuscript, RANKL and RANK are both detected in the epididymis, prostate and seminal vesicles. While the authors suggest a role for the RANK system in sperm maturation, i.e. acquisition of sperm motility and fertilizing ability, it is worth narrowing down the potential contribution of epididymal RANK system to male fertility control.

For instance, since RANK is present in elongated spermatid (Fig.1) does it remain on epididymal sperm? Is RANKL found secreted in the epididymal fluid?

Based on the reviewer's suggestion we have extended our investigations and provide more extended expression profiles of RANKL, RANK, and OPG in all epididymal sections and prostate from mice and humans in the revised manuscript. Moreover, we show presence of RANKL and tested the effects of denosumab on human spermatozoa and have extended the link between RANKL in seminal fluid and reproductive function. Novel Data are presented in Figures 4 and 5 and in supplementary material and described in results section.

Could RANKL/OPG ratio (e.g. following OPG treatment) modulate sperm motility and acrosome reaction?

We agree with the reviewer that this is important to examine whether Denosumab despite being an antibody enters the seminal fluid. Amgen has elegantly shown this (now cited in the discussion) but Denosumab was unable to induce any change in intracellular calcium concentration of human sperm and had no effect on the acrosome reaction when exposed to human spermatozoa (Figure 4).

Since Denosumab treatment increases progressive sperm motility in a category of infertile patients (Fig. 5D), a complementary mechanistic insight would be needed to better stratify the dichotomized responses observed following patient treatment.

We agree with the reviewer that it is likely that Denosumab may exert an effect on both testis (sperm count) and on the epididymis/prostate (motility and maturation) but it is beyond the scope of this manuscript to study in greater detail because it would require human studies as there are many differences in sperm physiology between mice and humans. We have provided some observational human data by showing associations between seminal fluid RANKL levels and semen quality and demonstrated that seminal RANKL levels decline substantially following Denosumab treatment which exerts some stimulatory effects on sperm production (gonadal) and motility (epididymal).

2- As stated in the introduction, OPG presents with off-targets and a short half-life (1.80). The authors should justify the use of OPG to treat mouse animals instead of Denosumab.

We have in line with this suggestion now stated this in the discussion, p. 12, l292: "Effects of Denosumab cannot be examined in rodent models because it only blocks RANKL in higher primates²⁷".

3- I.150 "might be the result of reduced apoptosis". The authors should include the quantification of apoptosis following OPG treatment by TUNEL assay (or equivalent approach) to strengthen what was observed ex vivo (Fig 3DE) and to corroborate this hypothesis.

We agree with the reviewer that all effects ideally should be shown by using several approaches. Initially, we focused intensely on germ cell proliferation and used most of the sparse tissue for BrdU incorporation as shown in Supplementary figure 6. There was a trend towards increased germ cell proliferation following treatment with Denosumab or OPG, but none of the treatment effects were statistically significantly different from vehicle treatment despite using testis tissue from 8 different patients. Instead the number of cleaved PARP positive cells and thus the number of apoptotic cells (clearly validated as the best quantifiable marker in this *ex vivo* model) were lower not just for Denosumab treatment but also for OPG treatment compared with vehicle. Combined, this clearly indicates that Denosumab and OPG influence germ cell survival. Unfortunately, we did not have enough tissue to support the effect on apoptosis using yet another method such as TUNEL. The impact of both Denosumab and OPG suggests that this effect is mediated by RANKL and not TRAIL as Denosumab does not block TRAIL.

4- I.96 "RANK...with more prominent expression in spermatogonia". There is a discrepancy between the results obtained by IF and IHC for RANK. For instance, while IF demonstrates a predominant expression in elongated spermatids (Fig 1A), IHC shows a strong/almost exclusive expression in spermatogonia (Sup Fig1). Since the antibodies used for these two technics do not recognize the same epitope (C-term vs. N-term), the authors should discuss if a cleavage of RANK could explain the detection of two forms of the protein. If this is the case an additional level of regulation of the RANK/RANKL system could be envisioned in the testis, as it has been observed in the intestine (Zhao et al, FEBS letter, 2013 <https://febs.onlinelibrary.wiley.com/doi/full/10.1016/j.febslet.2013.08.005>).

We agree with the reviewer that RANK may be cleaved and released. However, the antibodies used targets 1) an internal region of the protein (aa 261-379) not the N-terminal, and 2) the C-terminal end (aa 317-616). Since both antibodies target the same domain, it is unlikely that the discrepancy is caused by released versus membrane-bound RANK because none of the antibodies target the N-terminal part of the protein. We have now extended the fluorescence expression in both mice and humans, which shows presence of RANK in both spermatogonia and spermatids. Based on the IHC expression in mice and humans – it may be more accurate to state that expression is stronger in spermatogonia, but detectable also in the spermatids. Therefore, in accordance with fluorescence data we have changed the results section accordingly line 98: “RANK was expressed in the cytoplasm and membrane of all VASA-positive germ cells, while IHC showed the most prominent RANK expression in spermatogonia (Fig. 1B, 1D; Fig. S1A).”

5- Fig 2. The authors should specify what they mean by WT? Are the global and Sertoli-cell specific knockouts compared to C57BL/6 mice? To make sure that no unspecific phenotype is associated with either the presence of the recombinase itself or the floxed alleles, comparative controls should include Vasa:Cre, Amh:Cre, and rank1f/fl mice.

We agree with the reviewer and apologize for the confusion created. We have used the appropriate controls and have now changed the names to be more precise and specific as requested by both Reviewer 1 and 2. See answer to major point three by the Reviewer 1 for a detailed answer.

6-I.154 "OPG treatment of Sertoli-cell RANKL deficient mice did not increase fertility, while OPG treatment in global RANKL-deficient mice suppressed fertility all together and caused no pregnancies and litter size were zero in all six females". Please rephrase and discuss this result.

We agree that this point is of interest because inhibition of RANKL with an exogenous RANKL inhibitor appears to be detrimental or at least non-beneficial when RANKL activity is low. We have not been able to clarify the exact mechanism of action but the animal data are in line with the human study showing that high OPG (if the high OPG levels may be a response to the presence of high RANKL levels) is a predictor of a beneficial response while low activity is predictive for a poor response. We have now extended the discussion to include this putative link between rodent and human data.

7. Fig 4. Correlative studies are performed on a global cohort of "infertile patients" vs. healthy men. While aetiology of male infertility is not always explained, the authors should stratify their cohort (i.e. asthenozoospermic, azoospermic, oligozoospermic patients) to perform additional correlations. This would help further discriminate the association of RANKL with sperm production and/or post-testicular maturation processes.

Based on this suggestion, we have conducted analyses in asthenozoospermic, azoospermic, and oligozoospermic patients and show no major differences in the associations between these subgroups although we lose power compared with the original analysis. In general, the associations were stronger in men with normal sperm counts and motility probably due to larger sample size. We have not included this in the supplementary material because we feel the data load is substantial and the observed associations provide

limited additional information.

Minor comments:

1- I.93. "RANKL, RANK and OPG were detected at transcriptional and protein level". While protein detection somehow implies a transcriptional expression, there is no gene expression analysis shown in the mouse. Please remove "transcriptional" from the sentence.

We have removed it from the sentence.

2- I.372. Please define TGCT. We have now defined TGCT p. 3, l 65-66: "in testicular germ cell tumours (TGCTs)"

3. I.106. Please define OPG-FC

We have now defined this as osteoprotegerin–immunoglobulin Fc segment complex.

Reviewer #3 (Remarks to the Author):

Male infertility is a common condition with limited treatment options. The authors present a highly novel set of experiments investigating the role of RANKL as a regulator of male infertility. The paper is well written. The results create an overall convincing story which would have broad interest to the scientific community. I have some comments which require attention:

1. Many of the results are highly variable, as is usual especially with sperm parameters. Please present as mean +/-SD rather than SEM.

We agree with the reviewer that semen quality parameters are highly variable and that the use of SD rather than SEM is a worthwhile and relevant discussion. Throughout most of the manuscript we have provided individual data points to show interindividual distribution within each group/genotype and in the human studies each man is represented by dots or lines. In most statistical models we compare data using ANOVA with Dunnett's test to adjust for multiple comparisons – so we believe it is a better choice to present data as mean with SEM or 95 CI (1.96*sem) instead of SD. SD illustrates the distribution, but the distribution is already clearly

visible by using dot plots, or as shown in the pilot intervention study – lines for each individual participant. Providing mean with SEM gives a quick overview of the significant change over time, which seems to be the relevant output here.

2. It is important to present serum testosterone levels for the mouse KO models, since testicular steroidogenesis will inevitably affect spermatogenesis. Without this, it is difficult to interpret the results.

We agree with the reviewer and show the serum testosterone levels (measured by LC-MS/MS) in all mice models (Figure 2I) and provide also serum testosterone levels in both the human association and intervention study (Figure 5 and Supplementary figure S7).

3. Fig 4. I think it is important to be clear that the observed correlations between semen RANKL and other parameters are weak. Please discuss and speculate why.

We agree with the reviewer that other factors for instance testosterone and FSH may be more important for semen quality than RANKL. We have therefore extended the discussion and highlighted this point in the discussion line 329: “The associations between semen quality variables and seminal RANKL levels were not strong, which implies that other regulators such as FSH and testosterone likely exert a stronger regulatory effect on spermatogenesis and sperm maturation than seminal RANKL concentration”

4. Fig 4. Please also include correlation between semen RANKL and serum testosterone.

We agree and have now included this with estradiol and testosterone/LH in figure 5G and in the results section and discussion.

5. Fig 5. Given the small number of participants, I think post-hoc analysis of subgroup responses is highly suspect. I take your point that high OPG MAY signify likely response to therapy, but this clearly requires prospective, dedicated confirmation. Please highlight in discussion. Furthermore, please be explicit in the abstract that there was no change in sperm parameters in the recruited cohort of men with infertility.

We agree with the reviewer that the human intervention should be considered a pilot study and have now changed this accordingly in the abstract, subheading, result section and discussion in the revised manuscript.

We appreciate the constructive review and hope you will find the revised version suitable for publication in Nature communications.

All authors accept responsibility for the content of this manuscript.

Kind regards,

Martin Blomberg Jensen

REVIEWER COMMENTS

Reviewer #1 (Remarks to the Author):

The manuscript has improved considerably and the authors have tried to answer the reviewers' questions to the best of their ability. Nevertheless, there are several points where some technical details remain to be clarified. In particular

1) Fertility tests. The description is much more complete and allows a better understanding of the design of the experiments. However, several points still need to be clarified or improved. First, it is important to mention how many males were tested for each genotype. Secondly, I am still not convinced by Figure 2J, in particular the first two graphs on (i) the pregnancy rate and (ii) the average number of pups per female. It would be more relevant to replace them with (i) plug latency (i.e. the average number of days for the female to be plugged), (ii) the impregnation rate in % representing the % of gestation rate after a female is plugged and (iii) litter size. What seems clear is that once a female is fertilized, the size of the litter is not affected by the genotype. The variation in the pregnancy rate could therefore be explained by defects in mating behavior. Alteration in mating behavior can be assessed simply by monitoring (i) grooming with anogenital investigation, (ii) mounting with pelvic thrusts and (iii) ejaculation as revealed by the presence of a vaginal plug (see Crawley, J. N. (2007) *What's Wrong With My Mouse?* John Wiley & Sons, Inc., Hoboken, NJ, USA).

2) Seminal concentrations of RANKL are positively associated with serum estradiol. Is there an association between seminal RANKL, serum estrogen levels and semen quality? In my opinion, it would be worth putting this aspect into perspective in the discussion.

Minor details:

- 1) Legend figure 2E: what means ABC sperm motility
- 2) Legend figure 2J: "pups" instead of "pubs"
- 3) Legend Figure 4D: how many replicates? It should be at least n=5

Reviewer #2 (Remarks to the Author):

The authors addressed my main questions.

Reviewer #3 (Remarks to the Author):

The authors have considered and provided appropriate responses to address my comments.

Reviewer #4 (Remarks to the Author):

The authors showed the novel significance of the RANKL/RANK/OPG system in male reproductive function. In human studies, the authors found higher seminal sRANKL concentration in infertile men, and negative association between seminal sRANKL concentrations and semen quality variables. Furthermore, it is found that men with high OPG level had an increase in total motile sperm after denosumb treatment. The findings in human studies and logics for the story seem to be interesting. However, there are a number of critical concerns in the experimental data and the interpretation is often biased. In particular, histological data and western blotting data on RANKL, RANK and OPG expression are very premature and confusing. Furthermore, the authors should make more efforts to provide mechanistic insights how RANKL regulate spermatogenesis and sperm maturation in the testis, epididymis and prostate. The revised MS is still immature and descriptive. There are a lot of concerns as follows.

1. The histological data on RANKL, RANK and OPG expression (Fig.1A-D) are very premature, and there are many concerns. It is so difficult to accurately distinguish spermatocytes, spermatids and spermatogonia for evaluation of RANKL expression in Fig.1A. The authors stated "RANKL was expressed in the cytoplasm/membrane of mature Sertoli cells... but not in spermatogonia", but RANKL seems not to be expressed in spermatids. In addition, Fig.1B clearly showed that RANK was highly expressed in the elongated spermatids, but Fig.S1A showed no positive signal of RANK in spermatids. The authors should clearly address this inconsistency, which makes the reviewer doubt if the antibodies work. The current data on the expression of RANKL, RANK and OPG in the mouse testis are unconvincing and insufficient to support the conclusion. The authors should describe the expression pattern of RANKL, RANK and OPG precisely based on the data.

2. There are also many concerns in the histology of human samples (Fig.3B-E). In Fig.3B, RANKL seemed to be expressed in some of spermatogonia (SOX9-negative cells). RANKL expression was clearly detected in spermatocytes in the mouse testis (Fig.1A), whereas it was hardly detected in human spermatocytes. OPG was expressed in spermatogonia, spermatocyte and spermatids in mice, whereas it was mainly expressed in spermatogonia in human. The authors should address these inconsistencies. Right micrographs are not identical to small white boxes in left micrographs in Fig3E.

3. Although the authors described in the abstract "RANKL signals through its receptor RANK expressed in spermatogonia", there was no experimental evidence that RANKL directly acts on spermatogonia via RANK. Is RANK expression on other germ cells such as spermatocytes and spermatids not important?

4. In Fig.3F, denosumab or OPG treatment seemed to inhibit the apoptosis of not only spermatogonia but also other germ cells. Again, the authors should clarify whether RANK on other germ cells such as spermatocytes is dispensable or not. It is necessary to elucidate which cells RANKL acts on in the testis. It is also required to elucidate the molecular mechanism how RANKL induces cell apoptosis or blocks cell cycle.

5. Considering the fact that 50% of Rankl fl/fl mice obtained when using Vasa-Cre mice had heterozygous loss of the Rankl gene as the authors mentioned, the Rankl fl/fl mice should not be used as a control. In order to clarify the significance of Sertoli cells as a source of RANKL in the testis, it is necessary to compare RANKL expression in the testis between Sertoli-specific RANKL-deficient mice and controls. The authors should show data only from flox/wt mice, or control littermates of Sertoli-specific RANKL-deficient mice as a control.

6. There were very critical concerns in Fig.2C. Both membrane-bound (<40kDa) and soluble (approximately 25kDa) RANKL was clearly detected even in Rankl^{-/-} lysates, indicating that the RANKL gene was not deleted in Rankl^{-/-} mice used in this study. Thus, the reliability of all the mouse data in the study is doubtful. In the blot using RANKL extrac antibody, why were two bands of membrane-bound RANKL detected? Upper is non-specific band? Based on Fig.2C showing that RANKL expression was not decreased in Sertoli-specific RANKL-deficient mice, it is unlikely that Sertoli cell is a crucial source of RANKL in the testis. Fig.2C made all the conclusions of the MS questionable.

7. As for Comment 5, the authors described in the main text "Phenotypic characteristics, including loss of tooth...consistent with phenotypic expectations of a global RANKL knockout model". The authors should provide the data that Rankl^{-/-} (RANKL fl/fl Vasa-Cre) mice exhibited severe osteopetrosis. In page 6 line 125, "bone weight" should be changed to "bone mass".

8. Why was testicular and epididymal weight not so high in Rankl^{-/-} mice as in Sertoli-specific RANKL-deficient mice? On the other hand, the pregnancy rate was high in Rankl^{-/-} mice, indicating that the other organ-derived RANKL is the most important for male reproduction. There are many discrepancies in Fig.2.

9. It is still unclear why OPG treatment in Rankl^{-/-} mice completely suppressed fertility. These data suggest the possibility that OPG treatment affects male reproductive function in a RANKL-independent manner. Unless the authors clearly address the issue, all the data regarding OPG treatment are meaningless.

10. The authors also showed RANKL and RANK expression in the epididymis and prostate, but it is totally unclear how the RANKL-RANK interaction in the epididymis and prostate is involved in sperm maturation. Which cells did RANKL act on in the epididymis and prostate? As commented above, the authors described that RANK was prominently expressed in spermatogonia in mouse and human (Fig.1 and 3). Nevertheless, the authors implied in Discussion section the role of RANK on late spermatids in sperm maturation. Throughout the MS, it is largely unclear which cell-derived RANKL acts on RANK on which cell in the testis, epididymis and prostate.

11. It is also still unclear why denosumab suppress RANKL production in the seminal fluid and in vitro culture media of testis tissue.

Minor points

1. In page 3 line 74, "immature osteoclasts" should be changed to "osteoclast precursor cells".

2. In the introduction section, the authors should precisely describe the information of soluble RANKL and RANKL reverse signaling on the basis of the scientific evidence and cite the appropriate literatures. In page 3 line 78, refs 19-21 are not related to soluble RANKL function in glucose homeostasis. In page 3 line 79, "soluble RANK" is somewhat misleading. "Vesicular RANK" is better. The authors should cite the original paper (Ikebuchi, Nature, 2018) as the reference on RANKL reverse signaling.

Reviewer #1 (Remarks to the Author):

The manuscript has improved considerably and the authors have tried to answer the reviewers' questions to the best of their ability. Nevertheless, there are several points where some technical details remain to be clarified. In particular

1) Fertility tests. The description is much more complete and allows a better understanding of the design of the experiments. However, several points still need to be clarified or improved. First, it is important to mention how many males were tested for each genotype. Secondly, I am still not convinced by Figure 2J, in particular the first two graphs on (i) the pregnancy rate and (ii) the average number of pups per female. It would be more relevant to replace them with (i) plug latency (i.e. the average number of days for the female to be plugged), (ii) the impregnation rate in % representing the % of gestation rate after a female is plugged and (iii) litter size. What seems clear is that once a female is fertilized, the size of the litter is not affected by the genotype. The variation in the pregnancy rate could therefore be explained by defects in mating behavior. Alteration in mating behavior can be assessed simply by monitoring (i) grooming with anogenital investigation, (ii) mounting with pelvic thrusts and (iii) ejaculation as revealed by the presence of a vaginal plug (see Crawley, J. N. (2007) *What's Wrong With My Mouse?* John Wiley & Sons, Inc., Hoboken, NJ, USA).

We have based on this suggestion included exact data and provide the number of males and females used for each genotype and the number of vaginal plugs during the controlled fertility study in the figure legend for figure 2 and in the results section. We agree with the reviewer that vaginal plugs in theory would be a good marker for ejaculation and the fertilizing ability of the sperm could be determined based on actual pregnancies. However, in 15% of all pregnancies no plugs were observed. This challenges the use of the proposed impregnation rate and can therefore not be considered a robust and valid measurement as it is unlikely that the observed frequency of plugs is precise when 15% of pregnancies occur without a single plug despite 10 assessments in 5 days. However, we agree with the reviewer that the data on vaginal plugs provide the best available information about sexual activity, and we have therefore included a new section line 157: "The differences in fertility rates between genotypes were in part mirrored by number of days with verified plugs in exposed females (Rankl fl/fl 39%, Rankl fl/fl + OPG 56%, SC-Rankl 58%, SC-Rankl + OPG 67%, Rankl -/- 83% and Rankl -/- + OPG 50%). However, the high frequency of verified plugs in OPG treated *Rankl* -/- that produced no litters and the pregnancies without verified plugs in 15% of all females challenge the conclusions drawn from this observation.", and in the discussion line 305: "Mice with global deficiency of RANKL has increased male fertility but the exact mechanism of action was not demonstrated as the increase in sperm production and sexual activity were not statistically significantly different from control mice." and line 320: "Interestingly, the observed compensatory low testicular OPG expression in mice with global RANKL deficiency may facilitate increased TRAIL mediated apoptosis. This could explain why there was no change in testicular weight compared with control mice while the increase in testicular weight in Sertoli RANKL deficient mice may be due to no compensatory change in testicular OPG and TRAIL signaling. Injection of OPG into global RANKL deficient mice suppressed fertility despite of normal sexual activity, which indicates that the effect on fertility of treatment with a RANKL inhibitor may depend on presence of testicular RANKL and OPG activity."

2) Seminal concentrations of RANKL are positively associated with serum estradiol. Is there an association between seminal RANKL, serum estrogen levels and semen quality? In my opinion, it would be worth putting this aspect into perspective in the discussion.

We agree with the reviewer that the link between sex steroids and seminal RANKL levels are of interest and that RANKL may be regulated by estradiol and testosterone. We have extended and rephrased a section in the discussion starting line 345: “The 100-fold higher sRANKL concentration in seminal fluid compared with serum supports a biological role. It is plausible that RANKL is secreted, cleaved, or released from the epithelial cells in the reproductive tract under the control of local regulators as there was no link between serum and seminal concentrations of sRANKL. A biological role of seminal RANKL was indicated by the negative associations between seminal sRANKL concentrations and all semen quality variables including the total number of progressive motile sperm and total number of morphological normal sperm that refers to the number of mature and functional sperm and not just quantity. The strong associations between seminal RANKL levels and serum testosterone and estradiol suggest that RANKL activity and release into the epididymis and prostate may be influenced by sex steroids. The associations between semen quality variables and seminal RANKL levels were not strong, which implies that other regulators such as FSH, estradiol, and testosterone likely exert a stronger regulatory effect on spermatogenesis and sperm maturation than seminal RANKL concentration.”

Minor details:

1) Legend figure 2E: what means ABC sperm motility

it means sperm motility and we have removed ABC –in humans it is used to distinguish sperm motility (ABC) from progressive sperm motility (AB).

2) Legend figure 2J: “pups” instead of “pubs” **changed**

3) Legend Figure 4D: how many replicates? It should be at least n=5 **repeated twice in three different donors**

Reviewer #2 (Remarks to the Author):

The authors addressed my main questions.

Reviewer #3 (Remarks to the Author):

The authors have considered and provided appropriate responses to address my comments.

Reviewer #4 (Remarks to the Author):

The authors showed the novel significance of the RANKL/RANK/OPG system in male reproductive function. In human studies, the authors found higher seminal sRANKL concentration in infertile men, and negative association between seminal sRANKL concentrations and semen quality variables. Furthermore, it is found that men with high OPG level had an increase in total motile sperm after denosumb treatment. The findings in human studies and logics for the story seem to be interesting. However, there are a number of critical concerns in the experimental data and the interpretation is often biased. In particular, histological data and western blotting data on RANKL, RANK and OPG expression are very premature and confusing. Furthermore, the authors should make more efforts to provide mechanistic insights how RANKL regulate spermatogenesis and sperm maturation in the testis, epididymis and prostate. The revised MS is still immature and descriptive. There are a lot of concerns as follows.

1. The histological data on RANKL, RANK and OPG expression (Fig.1A-D) are very premature, and there are many concerns. It is so difficult to accurately distinguish spermatocytes, spermatids and spermatogonia for evaluation of RANKL expression in Fig.1A. The authors stated “RANKL was expressed in the cytoplasm/membrane of mature Sertoli cells... but not in spermatogonia”, but RANKL seems not to be expressed in spermatids. In addition, Fig.1B clearly showed that RANK was highly expressed in the elongated spermatids, but Fig.S1A showed no positive signal of RANK in spermatids. The authors should clearly address this inconsistency, which makes the reviewer doubt if the antibodies work. The current data on the expression of RANKL, RANK and OPG in the mouse testis are unconvincing and insufficient to support the conclusion. The authors should describe the expression pattern of RANKL, RANK and OPG precisely based on the data.

We agree with the reviewer that the description and presentation of the expression profile of RANKL, RANK, and OPG in the mouse testis were not optimal and have now revised this extensively and provide both novel figures and have changed the results section accordingly. The main discrepancy between the presented protein expression in the mouse (IF and IHC) and the human testis was due to use of different fixative and methodology. The originally presented mouse IF in Figure 1 of the version commented by the reviewer was performed on perfusion-fixed material, while the human testis were fixed in modified Stieve’s fixative and the mouse IHC presented in supplementary figure 1A was performed on formalin-fixed tissue. We have now conducted IF on formalin-fixed mouse testis and have uploaded an improved revised Figure 1A-D as shown below.

Accordingly, we have changed the description in the result section and line 99 now states: “RANKL was strongly expressed in the cytoplasm/membrane of mature Sertoli cells co-expressing SOX9 in the nucleus. RANKL expression was also detected in the cytoplasm of most spermatocytes and some spermatids but not in spermatogonia (Fig. 1A,1D; Fig. S1A). RANK was expressed in the cytoplasm and membrane of the VASA-positive germ cells, particularly spermatogonia and spermatids and in some spermatocytes (Fig. 1B, 1D; Fig. S1A). OPG was expressed in the cytoplasm of most spermatogonia, the junction between peritubular cells and spermatogonia, spermatids, and some peritubular cells (Fig. 1C-D).”

The use of formalin fixed testicular tissue results in sub-optimal morphology, but we can now directly compare the cellular expression pattern found in the mouse to the human testis as we have included novel IF stainings on formalin-fixed testis tissue in supplementary figure 5. This information is described in great detail in the results section in the revised manuscript – see also answer below to comment 2.

2. There are also many concerns in the histology of human samples (Fig.3B-E). In Fig.3B, RANKL seemed to be expressed in some of spermatogonia (SOX9-negative cells). RANKL expression was clearly detected in spermatocytes in the mouse testis (Fig.1A), whereas it was hardly detected in

human spermatocytes. OPG was expressed in spermatogonia, spermatocyte and spermatids in mice, whereas it was mainly expressed in spermatogonia in human. The authors should address these inconsistencies. Right micrographs are not identical to small white boxes in left micrographs in Fig3E.

We apologize for the understandable confusion created by the used setup. We have now used the same fixative and methodology in the presented mouse IF in Figure 1A-D and human IF in supplementary Figure 5. Moreover, we have improved figure 3B-D. Therefore, we can now extrapolate the expression pattern observed in the mouse to the human expression (since testis tissue from both species are fixed in formalin). The discrepancy was largely due to fixation since formalin-fixed human testis show virtually similar expression of RANKL, RANK, and OPG as shown for mice in Figure 1. In the revised Figure 3B-D, the human testis tissue is fixed in modified Stieve's fixative to show more appropriate testicular morphology. When using this fixative, we can corroborate the main expression pattern observed in mouse and formalin fixed tissue with Sertoli-spermatogonia/spermatocyte interaction. The description of these findings have been included in the revised manuscript line 177:" The expression pattern of RANKL, RANK, and OPG was evaluated in formalin-fixed human testis to extrapolate the findings from mice to humans (Fig S5A-E). By using antibodies targeting the transmembrane or extracellular domain we found cytoplasmic expression of RANKL in the SOX9-positive Sertoli cells and in some of the germ cells on the luminal site of the blood-testis barrier (Fig. S4B; Fig. S5A,S5D-E; Table S1). RANKL expression in germ cells was detected primarily in, in some spermatocytes and spermatids but not in spermatogonia (Fig. S4B; Fig. S5A,-S5D-E; Table S1). RANK was expressed in the cytoplasm/membrane of the germ cells with the most prominent expression in spermatogonia and spermatids (Fig. S4B; Fig. S5B-E; Table S1). OPG was expressed in spermatogonia and less frequently in peritubular cells or at the border between the peritubular cells and spermatogonia and in a fraction of spermatids (Fig. S4B; Fig. S5C-E; Table S1). The same antibodies were also applied on human testis fixed in modified Stieve's solution to improve testis morphology. Here, RANKL expression was detected mainly in the cytoplasm/membrane of the Sertoli cells and more rarely in spermatocytes and spermatids (Fig. 3B, 3E; Table S1). RANK was expressed in the cytoplasm/membrane of the germ cells with the most prominent expression in spermatogonia and spermatids (Fig. 3C, 3E; Table S1), while OPG was strongly expressed in spermatogonia, some of the peritubular cells, the border between the peritubular cells and spermatogonia and in many spermatocytes and spermatids (Fig. 3D-E; Table S1).."

RANKL expression was not detected in any spermatogonia, but for some Sertoli cells the nuclear SOX9 expression may be present in the "next" layer if the nucleus is residing deeper within the tissue block. When the majority of Sertoli cells are in the same layer it is obvious that RANKL is confined to the Sertoli cells as shown below in the mouse testis (Mouse – Sertoli-germ cells) or when co-stained with a spermatogonia marker MAGE-A4 in human testis (Supp figure S5A). The figure below shows RANKL and RANK with no overlap and with most Sertoli cells in the same layer of mouse testis

The figure below shows RANKL, SOX9 (Sertoli), and MAGE-A4 (spermatogonia) expression in human testis with no overlap in expression between RANKL in the SOX9-positive Sertoli cells and the MAGE-A4-positive spermatogonia.

RANK expression appeared less intense in spermatocytes in the formalin-fixed testis compared with modified Stieve's fixative, but it is expressed regardless of the used fixative. OPG expression seemed also to be comparable between species as shown in revised Figure 1 and supplementary Figure 5, although some of the intensity of the expression in the spermatids and spermatocytes was lost in human testis fixed in modified Stieve's as shown in the previous and the improved and revised Figure 3.

Noteworthy, all the used antibodies have been validated in bone and breast tissue.

Revised Figure 3 BE

The novel Suppl. Figure 5 showing expression profile in formalin-fixed human testis.

3. Although the authors described in the abstract “RANKL signals through its receptor RANK expressed in spermatogonia”, there was no experimental evidence that RANKL directly acts on spermatogonia via RANK. Is RANK expression on other germ cells such as spermatocytes and spermatids not important?

Antibodies do not pass freely through the blood-testis barrier (BTB), which implies that less than 2% will enter so the vast majority of IgG antibodies like Denosumab will predominantly target the proximal site of the barrier where the Sertoli cells interact with the spermatogonia. However, the reviewer may refer to the understanding that Denosumab enters the seminiferous tubules (shown by labelled Denosumab in monkeys) and seminal fluid (in humans) referenced in the discussion, which supports that a small fraction enters and may block RANKL in the germ cells on the luminal site of BTB. Estimated 1-2 % of the maximum circulating fraction. The maximum concentration measured in the seminal fluid is

slightly higher than the 100 ng/ml dose used in the *ex vivo* culture so we actually use the “physiologically available concentration” in our model. It is difficult to test whether the Sertoli cells exclusively signal to the spermatogonia or if they also influence spermatocytes. This cannot be addressed without stage specific suppression of RANK and this is not easy in a human model system. The role of RANK in later stages of spermatogenesis may also be of importance especially since RANKL is also produced on the luminal site of the blood-testis barrier. Although it is more doubtful if the luminal RANKL-RANK interaction is blocked consistently by Denosumab treatment and therefore probably of less clinical importance. In conclusion, based on the expression profile of RANKL, RANK, and OPG and the physiology of the blood-testis barrier the best supported rationale is that Denosumab blocks the Sertoli-spermatogonia interaction. Still we have softened the conclusion and now write in line 430: “In conclusion, this translational study suggests that RANKL signaling is a novel regulator of male reproductive function by being a mediator in the Sertoli-Germ cell interaction in the male reproductive tract and thereby exerting an influence on semen quality and male fertility.”

4. In Fig.3F, denosumab or OPG treatment seemed to inhibit the apoptosis of not only spermatogonia but also other germ cells. Again, the authors should clarify whether RANK on other germ cells such as spermatocytes is dispensable or not. It is necessary to elucidate which cells RANKL acts on in the testis. It is also required to elucidate the molecular mechanism how RANKL induces cell apoptosis or blocks cell cycle.

We agree with the reviewer that this is an interesting question, which we hope we can address in future studies. We agree that the apoptotic cells often were spermatocytes, but tissue cultured *ex vivo* will result in cell death including apoptosis of all types of germ cells. In accordance, we also found apoptotic germ cells in the vehicle treated testis cultures. Still, the number of germ cells undergoing apoptosis was lower in the Denosumab treated cultures. However, to increase the possible relevance for repurposing we have provided more data related to predictive biomarkers for a beneficial response. Serum OPG may be a marker of systemic RANKL-OPG activity but we have now extended this data as we find it important to state that we do not believe that men with complete spermatogenic arrest of predominantly Sertoli cell only syndrome will benefit from RANKL inhibition. Now, we present data showing that Sertoli cell function/gonadal function (determined by Inhibin B and to some degree AMH) are predictive markers for the Denosumab response. RANKL is expressed in the Sertoli cells and is approachable (in high concentrations) by Denosumab and other inhibitors due to the presence on the proximal site of the blood-testis barrier. This may be highly relevant information for future RCTs testing the effect of Denosumab versus placebo in infertile men. The data have been included in Figure 6.

5. Considering the fact that 50% of Rankl fl/fl mice obtained when using Vasa-Cre mice had heterozygous loss of the Rankl gene as the authors mentioned, the Rankl fl/fl mice should not be used as a control. In order to clarify the significance of Sertoli cells as a source of RANKL in the testis, it is necessary to compare RANKL expression in the testis between Sertoli-specific RANKL-deficient mice and controls. The authors should show data only from flox/wt mice, or control littermates of Sertoli-specific RANKL-deficient mice as a control.

This is a valid point raised by the reviewer and we considered this issue years ago but decided to use littermate controls as this is the only appropriate control. We hope the reviewer will agree that by using littermate controls (some even with heterozygous loss of the Rankl gene) there will in theory be lower chances of showing significant differences compared with the Sertoli specific or systemic deficient models and that this cannot be considered a poor but rather conservative control. We believe our choice may

underestimate the effect of RANKL inhibition.

6. There were very critical concerns in Fig.2C. Both membrane-bound (<40kDa) and soluble (approximately 25kDa) RANKL was clearly detected even in Rankl^{-/-} lysates, indicating that the RANKL gene was not deleted in Rankl^{-/-} mice used in this study. Thus, the reliability of all the mouse data in the study is doubtful. In the blot using RANKL extrac antibody, why were two bands of membrane-bound RANKL detected? Upper is non-specific band? Based on Fig.2C showing that RANKL expression was not decreased in Sertoli-specific RANKL-deficient mice, it is unlikely that Sertoli cell is a crucial source of RANKL in the testis. Fig.2C made all the conclusions of the MS questionable.

We respectfully disagree with the reviewer on this point. Both mouse models are deficient models and not knockout models which is consistent with the repressed RANKL protein expression shown in Figure 2C. Importantly, in the Sertoli-specific deficient model the reduction of RANKL in Sertoli cells may be compensated by increased RANKL expression in the germ cells. However, in the systemic RANKL deficient mice where there is no RANKL the only compensation available is the observed decrease in the testicular expression of OPG. Interestingly, this decrease in testicular OPG cannot be found in the Sertoli-deficient model. This clearly indicates that loss of RANKL in all testicular cells enforces a larger need for compensation not on receptor level as RANK is unchanged but low OPG and thereby reduced RANKL and TRAIL inhibition. This supports that the actual RANKL deficiency in these mice are clearly distinct from the controls and the Sertoli deficient mice. All three RANKL isoforms are expressed in the mouse testis and may in theory be expressed in both Sertoli and germ cells. This alone or altered glycosylation and other posttranslational modifications may explain the additional band of the membrane bound RANKL.

7. As for Comment 5, the authors described in the main text “Phenotypic characteristics, including loss of tooth...consistent with phenotypic expectations of a global RANKL knockout model”. The authors should provide the data that Rankl^{-/-} (RANKL fl/fl Vasa-Cre) mice exhibited severe osteopetrosis. In page 6 line 125, “bone weight” should be changed to “bone mass”.

We agree with the reviewer and have changed ‘bone weight’ to ‘bone mass’. The increase in bone mass was also found in OPG treated mice thereby corroborating the systemic RANKL inhibitory effect. This has been added to the supplementary material as Fig. S3A.

8. Why was testicular and epididymal weight not so high in Rankl^{-/-} mice as in Sertoli-specific RANKL-deficient mice? On the other hand, the pregnancy rate was high in Rankl^{-/-} mice, indicating that the other organ-derived RANKL is the most important for male reproduction. There are many discrepancies in Fig.2.

We agree with the reviewer that there are discrepancies between the deficient models and have therefore extended the analysis and included more animals as shown in Figure 2D.

Subsequently, epididymal weight is higher in both *Rankl* deficient mice models, but testis weight remains elevated only in the Sertoli-specific RANKL deficient mice. The best explanation for this observation is that *Rankl* *-/-* mice have a compensatory decrease in testicular OPG that differs from the Sertoli deficient and control mice. OPG is a potent inhibitor of TRAIL that is also a potent stimulator of apoptosis in germ cells. (<https://www.nature.com/articles/1207232>, <https://link.springer.com/article/10.1007/s10495-006-0288-1>, <https://pubmed.ncbi.nlm.nih.gov/24736722/>,). Low OPG in the global RANKL deficient mice could lead to increased germ cell apoptosis and thereby explain why the testicular weight is unchanged. The increase in fertility despite of this may be explained by increased sperm survival and sperm motility (<https://rep.bioscientifica.com/view/journals/rep/148/2/191.xml>). We have extended the discussion to include these points in line 320: "Interestingly, the observed compensatory low testicular OPG expression in mice with global RANKL deficiency may facilitate increased TRAIL mediated apoptosis. This could explain why there was no change in testicular weight compared with control mice while the increase in testicular weight in Sertoli RANKL deficient mice may be due to no compensatory change in testicular OPG and TRAIL signaling. Injection of OPG into global RANKL deficient mice suppressed fertility despite of normal sexual activity, which indicates that the effect on fertility by using RANKL inhibition as a treatment option may depend on presence of testicular RANKL and OPG activity."

9. It is still unclear why OPG treatment in *Rankl* *-/-* mice completely suppressed fertility. These data suggest the possibility that OPG treatment affects male reproductive function in a RANKL-independent manner. Unless the authors clearly address the issue, all the data regarding OPG treatment are meaningless.

RANKL activity in the testis is together with serum OPG, Inhibin B, and AMH a key determinant for a beneficial response to RANKL inhibition. Men with high RANKL/OPG in seminal fluid have a more beneficial response to Denosumab supporting the idea that low RANKL activity (*Rankl* *-/-*) leads to a detrimental response as observed in the mice. We have extended and rephrased the discussion, particularly line 392: " Serum Inhibin B and AMH were also predictive markers for a beneficial response, which implies that men with poor gonadal function or low RANKL activity may have a detrimental response to Denosumab as supported by the reduced fertility following OPG treatment in RANKL deficient mice. The present study does not provide clinically valid information but biological insight by showing that Denosumab can lower seminal RANKL levels and improve reproductive function as demonstrated by increased serum levels of inhibin B and AMH as well as sperm output in the infertile men with the best gonadal function. Together, this corroborates the biological phenotypes shown in mice and the human *ex vivo* model."

10. The authors also showed RANKL and RANK expression in the epididymis and prostate, but it is totally unclear how the RANKL-RANK interaction in the epididymis and prostate is involved in sperm maturation. Which cells did RANKL act on in the epididymis and prostate? As commented above, the authors described that RANK was prominently expressed in spermatogonia in mouse and human (Fig.1 and 3). Nevertheless, the authors implied in Discussion section the role of RANK on late spermatids in sperm maturation. Throughout the MS, it is largely unclear which cell-derived RANKL acts on RANK on which cell in the testis, epididymis and prostate.

We agree with the reviewer that these are important questions to address. We plan to examine this in future studies, but it is likely that RANKL-RANK interaction may be implicated in sperm survival (as shown earlier for TRAIL/OPG (<https://rep.bioscientifica.com/view/journals/rep/148/2/191.xml>) and maybe even in sperm-oocyte binding. We have adjusted the discussion line 344: "This indicates that the possible role of

RANKL in human sperm may be different and could in theory be used for binding to the epithelia of the female reproductive tract or the cumulus cells surrounding the oocyte³⁶. The 100-fold higher sRANKL concentration in seminal fluid compared with serum supports a biological role. It is plausible that RANKL is secreted, cleaved, or released from the epithelial cells in the reproductive tract under the control of local regulators (as there was no link between serum and seminal concentrations of sRANKL). A biological role of seminal RANKL was indicated by the negative associations between seminal sRANKL concentrations and all semen quality variables including the total number of progressive motile sperm and total number of morphological normal sperm that refers to the number of mature and functional sperm and not just quantity.”

11. It is also still unclear why denosumab suppress RANKL production in the seminal fluid and in vitro culture media of testis tissue.

We do not know whether Denosumab inhibits RANKL from being released into the seminal fluid or whether it enters the seminal fluid through testis, prostate, and epididymis to bind the RANKL available there. But our data show that Denosumab can inhibit RANKL activity and release in the target tissues at the concentration measured in seminal fluid from men after 60 mg injection sc as we and others also find in serum – so this is a general mechanism and not just in the male reproductive system.

Minor points

1. In page 3 line 74, “immature osteoclasts” should be changed to “osteoclast precursor cells”.

Thank you – have been changed.

2. In the introduction section, the authors should precisely describe the information of soluble RANKL and RANKL reverse signaling on the basis of the scientific evidence and cite the appropriate literatures. In page 3 line 78, refs 19-21 are not related to soluble RANKL function in glucose homeostasis. In page 3 line 79, “soluble RANK” is somewhat misleading. “Vesicular RANK” is better. The authors should cite the original paper (Ikebuchi, Nature, 2018) as the reference on RANKL reverse signaling.

We appreciate this comment and have adjusted some of the references based on the reviewers recommendations and revised the introduction line 77: “RANKL can also be found in serum, suggesting a putative endocrine function or non-skeletal actions for instance a role in bone and muscle health, menstrual cycle and glucose homeostasis (9, 19-23). Moreover, a recent study showed how complex RANKL signaling is as it was suggested that reverse RANKL signaling occurs when the receptor RANK becomes soluble and binds to RANKL thereby inducing downstream signaling in the RANKL expressing cell 24

We hope the revised manuscript will be satisfactory to justify publication of our translational study.

We thank you for your time and consideration of our manuscript

Roland Baron, Jorma Toppari, Beate Lanske, Anders Juul and Martin Blomberg Jensen

REVIEWER COMMENTS

Reviewer #1 (Remarks to the Author):

The authors addressed my main questions.

Reviewer #4 (Remarks to the Author):

Unfortunately, the authors could not fully address my comments, and there still remain a lot of critical concerns. The revised MS is still immature and descriptive, and full of critical errors that make the interpretation impossible. The authors should make more efforts to improve the MS.

1. (Regarding previous comment #1 and 2)

The authors replaced all the data of Fig.1ABCD and Fig.3BCD, added new Fig. S5ABCD and changed the conclusion of the cellular source of RANKL, RANK and OPG in the testis, but there are still discrepancies. Fig.1B clearly showed that RANK was highly expressed in the elongated spermatids, but Fig.S1A and Fig.S5BD showed no positive signal of RANK in spermatids. In the main text, the authors stated that OPG was expressed in spermatids, but OPG expression was hardly detected in Fig1A and Fig. S5. The expression pattern of OPG was so different among Fig.1A, Fig.3D and Fig. S5C. The expression pattern should not be changed by the difference of fixation.

2. (Regarding previous comment #4 and 10)

The authors failed to provide experimental data of the mechanism how RANKL induces germ cell apoptosis and how the RANKL-RANK interaction in the epididymis and prostate is involved in sperm maturation. It is largely unclear which cell-derived RANKL acts on RANK on which cell in the testis, epididymis and prostate. The authors should make more efforts.

3. (Regarding previous comment #5)

The reviewers' response is not acceptable. Based on the data of tail DNA genotyping, the authors should be able to remove the mice with heterozygous loss of the Rankl gene and show the data of only flox/wt mice.

4. (Regarding previous comment #6 and 7)

The reviewer previously asked why mRANKL and sRANKL were clearly detected in "global RANKL-deficient (Rankl^{-/-})" mice. The authors mentioned that the Rankl gene of two alleles was deleted in germ cells (fertilized eggs) in RANKL flox/flox Vasa-Cre mice. So, these mice should be "global RANKL-deficient mice", which means that the mice have "NO" RANKL. Indeed, Fig 2B showed RANKL mRNA expression was hardly detected in "global RANKL-deficient (Rankl^{-/-})" mice, but Fig. 2C clearly showed RANKL expression is not completely deleted. The amount of mRANKL protein was reduced by about half, and sRANKL was not changed in "global RANKL-deficient (Rankl^{-/-})" mice, indicating that these mice were not Rankl^{-/-} mice. Nevertheless, the authors mentioned "Phenotypic characteristics, including loss of tooth eruption, lactation deficit, and increased bone mass in the Vasa;Cre model were consistent with phenotypic expectations of a global RANKL knockout model" (page 5, line129). Thus, the reviewer requested the authors to provide the data that Rankl^{-/-} mice exhibited severe osteopetrosis at the previous review stage, but the authors only showed the trabecular weight but not other data. The authors should provide the data of bone mass (microCT or histology), loss of tooth eruption and lactation deficit. It is very critical point. The authors seem not to understand well the discrepancies.

Furthermore, the raw data of western blotting (excel file) clearly shows the molecular weight of sRANKL is different between the blot with Ab-9957 antibody and the blot with NBP-61813 antibody. Nevertheless, the authors show in main figure (Fig. 2C) as if they were the same. This should be strongly criticized.

Unless the authors correctly address all these concerns, the reliability of all the mouse data is doubtful.

5. (Regarding previous comment #8)

By increasing sample size, the result of epididymis weight has been changed. Anyway, the authors failed to clarify why testicular weight was not so high in "Rankl^{-/-} mice" as in Sertoli-specific RANKL-deficient mice. The authors mentioned the involvement of TRAIL-induced apoptosis, but did not provide any experimental evidence. It is unclear why OPG expression is reduced in "Rankl^{-/-} mice" but not Sertoli-specific RANKL-deficient mice, and why the pregnancy rate was increased in "Rankl^{-/-} mice".

6. (Regarding previous comment #9)

The authors could not address my comments. The authors claimed that RANKL inhibition increased male fertility and sperm count. Administration of OPG should have no additional effect in "global RANKL-deficient mice". Nevertheless, the Fig. 2J data that OPG treatment in RANKL^{-/-} mice completely suppressed fertility means that OPG modulated male reproductive function in a RANKL-independent manner. The issue concerns the main conclusion of the MS.

7. (minor comment)

In Supplemental material, line 156, "8-26: pups from Vasa-Cre + Rankl^{fl/fl}" may be incorrect, because lane 7-14 are pups from Amh-Cre + Rankl^{fl/fl}.

To the Editor of Nature Communications

Thank you for your thorough handling of our manuscript. We are happy that the Editors have asked us to revise the manuscript and have now implemented the requested changes and conducted the experiments necessary in order to answer the question raised by reviewer 1.

Below you will find our response in **bold** to the issues raised by the Editor and reviewers.

Thank you again for submitting your revised manuscript "RANKL is a novel regulator of male reproductive function" to Nature Communications. We have now received reports from 2 reviewers. Having considered all of the reviewer comments and your responses, we have decided that we need to see a revised manuscript clearly addressing the remaining points in the text, including further discussion of the expression pattern (please see comments from Reviewer #1 below) and inclusion of the phenotypic data supporting global RANKL deficiency.

Below you will find a detailed answer to reviewer 1.

Loss of tooth eruption and lactation deficit are characteristic findings in some of the RANKL knock out mice. We have as requested included the phenotypic data supporting global RANKL deficiency in the *Vasa*;Cre mouse strain line 127:" Phenotypic characteristics, including loss of tooth eruption, lactation deficit, and increased bone mass in the *Vasa*;Cre model were consistent with the expectations of a global RANKL knockout model¹⁹. An increase in bone mass was evident following a rough but systematic evaluation of tibial bone mass (Fig. S3A). An assessment of the breeding schemes showed that some offspring of female *Vasa*-Cre mice died due to lactation deficiency (appendix A). Moreover, a fraction of offspring from both female and male *Vasa*-Cre mice suffered from loss of tooth eruption (appendix B)."

EMAIL COMMENTS FROM REVIEWER #1

A notable discrepancy is the absence of expression of RANKL, and RANK in sertoli cell only testis (fig 3A) suggesting that the expression of these two proteins is germ cell specific while OPG is only partially reduced. This potential expression of OPG in the somatic part of the gonad is not really confirmed by the IFs (Fig 3D) which shows some labelling on spermatogonia. PTM labelling seems anecdotal to me and should not be able to explain these variations in OPG expression.

We agree with the reviewer that the observation of relatively high *OPG* expression in testicular specimens with Sertoli-cell-only tubules (SCO) is unlikely to be the result of peritubular expression only. However, the relatively high *OPG* in comparison with *RANKL* and *RANK* could be explained by the difference in cellularity between SCO and normal testis samples. Therefore, we conducted IHC on SCO specimens from four different patients and found virtually no RANK expression, and a modest RANKL expression in the Sertoli cells. Interestingly, *OPG* was markedly expressed in the cytoplasm of the Sertoli cells in SCO specimens. Images illustrating this expression pattern is included in Fig. S4D in the revised manuscript (and are included below).

The expression pattern of *OPG* in SCO is clearly distinct from the expression observed in the seminiferous tubules with "normal" Sertoli cells and presence of germ cells. It is well-recognized that the Sertoli cells in SCO tubules are different from Sertoli cells in tubules with germ cells, but the morphology and expression pattern can also differ substantially between SCO specimens.

The reviewer's suggestion has fostered a new discovery that needs further workup because our findings indicate that loss of germ cells or impaired Sertoli cell maturation facilitates OPG expression in the Sertoli cells. The biological implications for the expression of OPG in Sertoli rather than germ and peritubular cells are unknown. However, RANKL, RANK and OPG expression should be investigated throughout development to determine whether the change in expression is due to impaired maturation of the Sertoli cells or secondary to the germ cell loss. This novel finding supports the intervention data showing that the effect of Denosumab depend on AMH and Inhibin B and thus indirectly the etiology and severity of the male infertility.

This information has been included in Results and Discussion of the revised manuscript. Results Line 199: "Analyses of the expression in Sertoli-cell-only tubules from four patients showed no detectable RANK expression, but some cytoplasmic RANKL expression in the Sertoli cells characterized by SOX9 and Vimentin expression. Noteworthy, OPG expression was detected in the cytoplasm of the Sertoli cells and was thus strikingly different from the expression observed in testicular samples that contained seminiferous tubules with "normal" Sertoli cells and germ cells, where OPG was expressed in peritubular cells and germ cells (Fig. S4D)."

Discussion line 344: "Moreover, RANKL signaling is critically dependent on the etiology and severity of the male infertility because OPG was markedly expressed in the Sertoli cells in Sertoli-cell-only tubules and was thus, clearly different from the peritubular/germ cell expression found in normal seminiferous tubules. Future studies are needed to investigate RANKL signaling during development and in dysgenetic testes to determine whether the change in expression is the result of impaired Sertoli cell maturation or secondary to the germ cell loss."

Fig S4D

(D) IHC of RANKL (sc-7628), RANK (sc-9072), OPG (sc-8468) and Sertoli cell markers SOX9 (ab5535) and Vimentin (sc-373717) in tubules with Sertoli cell-only syndrome. Negative control is staining without addition of primary antibody. Neg. Ctrl, Negative control without primary antibody

We hope the revised manuscript will be satisfactory to justify publication of our translational study.

We thank you for your time and consideration of our manuscript

Roland Baron, Beate Lanske, Jorma Toppari, Anders Juul, Anne Jørgensen and Martin Blomberg Jensen